# Assessing the predictive capability of several machine learning algorithms to forecast snow avalanches using numerical weather prediction model in eastern Canada.

Francis Gauthier[1,2], Jacob Laliberté[1,3], and Francis Meloche[1,2]

[1]Laboratoire de géomorphologie et de gestion des risques en montagnes (LGGRM), Département de Biologie, Chimie et Géographie, Université du Québec à Rimouski, Canada.
[2]Center for Nordic studies, Université Laval, Québec, Canada.
[3]Horos Géomatique, Québec, Canada

**Correspondence:** Francis Gauthier (francis_gauthier@uqar.ca)

**Abstract.** Snow avalanches are a serious threat to traffic in the northern Gaspésie region. In this study, we look at the development of different forecasting models using machine learning (ML), based on snow avalanche events recorded by Québec's Ministry of Transportation, meteorological data from the Cap-Madeleine station and Environment Canada weather forecast data. The models were trained and tested on *Train* and *Test* datasets with meteorological and weather forecasts recorded at

the Meteorological Station. Unsupervised learning models were compared to expert models where only 4 variables were selected with avalanche expertise in mind, yielding similar results in prediction. The ML models were then tested in a realistic forecasting context over the year 2019 with weather data from a forecasting station (Hindcast) and with weather forecast data over 24h and 48h. The LR and RF models show that model performance can match or exceed that of current forecasting tools, enhancing hazard anticipation while maintaining a user-friendly framework suitable for real-time application. In conclusion,

recommendations on forecast-based operational procedures are proposed.

## 1 Introduction

Every year, of the 1.5 million potentially fatal avalanches in Canada, 5% occur in areas inhabited or frequented by humans. They are the most fatal winter hazard in the country (Hétu et al., 2015; Stethem et al., 2003). Between 2003 and 2020, the Québec's Ministry of Transportation (MTMQ) recorded more than 600 avalanches and 17 road accidents caused by avalanches on the

roads of the Gaspé Peninsula (Gauthier et al., 2022). Three people have been killed in road accidents caused by avalanches in this area (Fortin et al., 2011; Hétu, 2010; Hétu et al., 2011). Although the probability of an avalanche hitting a car or train directly remains relatively low (McClung, 1999), avalanches have a major socio-economic impact when major public transport routes are blocked by snow avalanche deposits, with the direct annual cost of highway closures exceeding 5 million Canadian dollars per year (Stethem et al., 2003; Jamieson and Stethem, 2002).

Since the start of the mass movement inventory program in 1987, MTMQ patrollers have been on the road around the clock (24 hours a day, 365 days a year) on the two roads (132 and 198) servicing the area. In the absence of an avalanche forecasting program, this reactive management approach needs to be supported by preventive management methods. There are various

approaches to prevent snow avalanches on transportation corridors. The first is obviously to avoid the hazard by moving roads away from the runout zone (e.g. Jaboyedoff and Labiouse, 2011; Michoud et al., 2012). In Gaspesia, slopes and avalanche

paths have been mapped at various scales (Germain, 2006; Royer and Lemieux, 2006). In many places, the road cannot be relocated because it is wedged between the slopes and the St. Lawrence estuary. In the late 90s, protective berms were erected by the MTMQ to reduce the number of mass movements reaching the roads. Despite the effectiveness of these infrastructures in limiting the runout of rockfalls in summer, they are less effective in countering avalanche runout in mid-winter and spring when the berms are filled with wind-hardened snow (Gauthier et al., 2017).

The required quality of meteorological data needed to feed physical snow simulation models (e.g. Morin et al., 2020) restricted us to statistical avalanche forecasting approaches based on the analysis of avalanche and meteorological data. The first step in improving data-driven avalanche forecasting is to establish causal links between meteorological conditions and their occurrence (Ancey, 2006; Castebrunet et al., 2012; Durand et al., 1999; Germain, 2016; Jomelli et al., 2007). A wide variety of statistical approaches have been used to explain and predict avalanche occurrence (e.g. Perla, 1970; Bois et al., 1975; Buser,

1983; Hendrikx et al., 2014). On a seasonal scale, logistic regression (LR) has been used to establish relationships between winter weather conditions and very large avalanches (Hebertson and Jenkins, 2003; Jomelli et al., 2007). This type of analysis has rarely been used to support the development of operational statistical forecasting models on a daily scale (Jomelli et al., 2007; Gauthier et al., 2017). Classification trees (CT) were also used to predict avalanche days according to a series of criteria or trigger thresholds (Davis et al., 1999; Hendrikx et al., 2005, 2014; Peitzsch et al., 2012; Gauthier et al., 2018). The use of

automated machine learning methods, such as neural networks (NN) or random forest (RF), is now more widely used to support the development of statistical avalanche forecasting models (e.g. Singh et al., 2005; Schirmer et al., 2009; Blagovechshenskiy et al., 2023). Recently, these more elaborate machine learning algorithms have been coupled with snow cover model outputs to predict avalanche danger level (Pérez-Guillén et al., 2022), wet avalanche activity (Hendrick et al., 2023), and dry-snow avalanche activity in the Alps (Mayer et al., 2023; Viallon-Galinier et al., 2023). While the implementation of snow cover

models is promising, these types of machine learning methods have rarely been tested with numerical weather prediction data (NWP) or in an operational context. Yet, only one study tested the performance of 24-hour NWP to predict wet snow avalanche activity (Hendrick et al., 2023). Ultimately, the machine learning method used remains a choice on which there is no consensus and sometimes seems to follow the major trends of the moment. Very few studies have evaluated the advantages and disadvantages (Davis and Elder, 1994) or compared the performance of these various statistical and machine learning methods

(Schirmer et al., 2009; Gauthier et al., 2017). Our study aims to train and test the performance of different forecasting models. We use four machine learning (ML) algorithms to predict avalanche days based on various meteorological variables. Our study also aims to evaluate and compare the predictive performance of each machine learning algorithm in an operational avalanche forecasting context using NWP data. We then discuss the link between the probability of an avalanche occurring and avalanche danger level, providing a more solid base for risk management operational procedures.

## 2   Study area

The study area is located on the north shore of the Gaspé Peninsula, between the towns of Sainte-Anne-des-Monts and Manche d'Épée (Figure 1). For almost its entire length (80 km), road 132 is hemmed in between the shoreline of the St. Lawrence estuary and the steep slopes that form the coastal escarpment, with all slopes facing north (0° ± 45°). According to Hétu (2010), Fortin et al. (2011) and Gauthier et al. (2017), slopes conducive to snow avalanches can be classified into three groups: 1) scree slopes below rock faces; 2) forest corridors; and 3) sparsely forested slopes. Along road 132, the majority of avalanches start from talus slopes. This is the case, for example, on section 100 west and east of Mont-Saint-Pierre (Figure 2ab) and on sections 120 and 130 between Gros-Morne and Manche d'Épée (Figure 2c). In some places, avalanches originate from heavy snow accumulations in landslide scars that form corridors in the forest canopy (Figure 2d). The lower slope was truncated over a distance of almost two kilometers to allow the road to be built. The steepened slopes form a convexity that increases the instability of the snowpack (McClung and Schaerer, 2006).

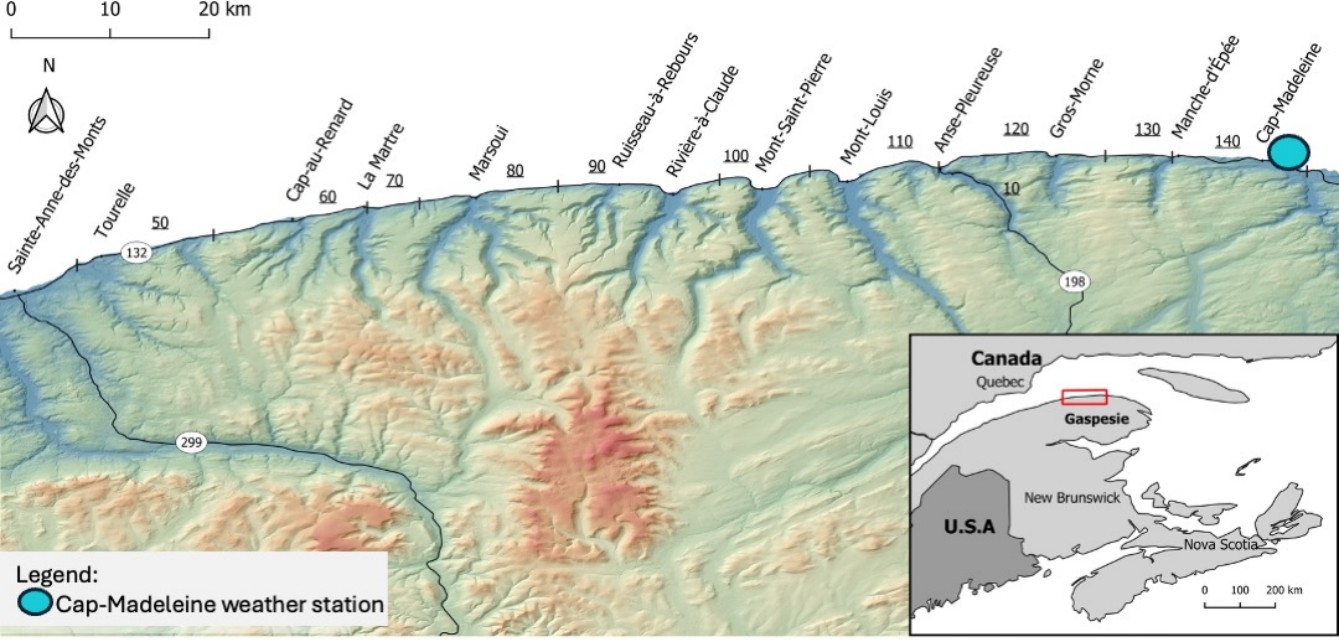

**Figure 1.** Location of study area and Cap-Madeleine weather station. Underlined numbers are sections of road 132.

The region is characterized by a humid continental climate with short, cool summers. The average annual temperature is 3.2 °C, with the hottest month averaging 16.5 °C (July) and the coldest month averaging -11.6 °C (January). Average annual precipitation is 864 mm, 28 % of which falls as snow (Environment Canada, 2022). The region's winter climate is characterized by an alternation of contrasting weather conditions: 1) continental lows originating from the North American Cordillera (Colorado and Alberta) or maritime lows originating from the Gulf of Mexico, accompanied by strong northeasterly

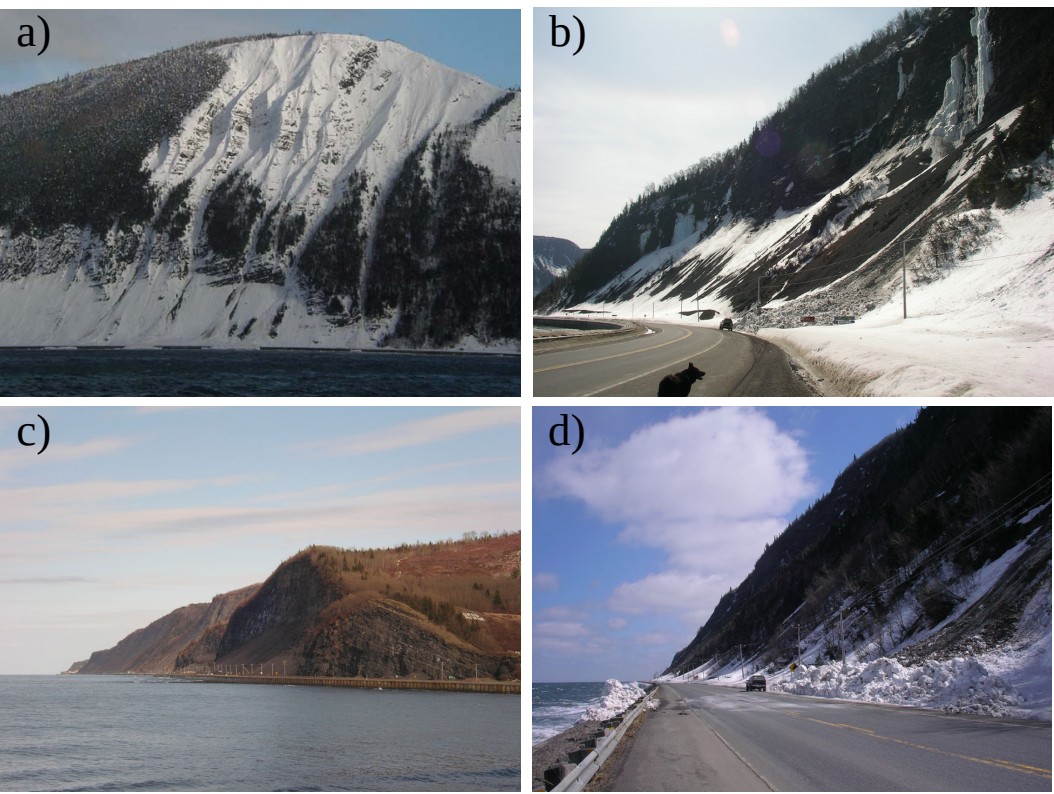

**Figure 2.** View of avalanche slopes along road 132 east (a) and west (b) of Mont-Saint-Pierre (section 10), along road 132 east of Gros-Morne (sections 120 and 130) (c and d).

winds (> 60 km/h), both bringing significant snow accumulations of up to 100 cm in 48 hours; 2) arctic air masses with strong northwesterly winds and temperatures below -20 °C; and 3) warm air masses (> 0 °C) from the southern United States, sometimes accompanied by rain (Meloche, 2019; Gauthier et al., 2017; Fortin and Hétu, 2009, 2014).

The North-Gaspésian climatic context is particularly conducive to storm avalanches and wet snow avalanches (Hétu, 2010; Fortin et al., 2011; Gauthier et al., 2017, 2018). Between 1987 and 2020, 861 snow avalanches spread over 153 event days were recorded by the MTMQ on road 132 (Figure 1). Considering that merlons were built during the 90s to limit the runout of rockfalls, ice-block falls, and avalanches, it is unlikely that any avalanches reached the road in 10 out of 17 years before winter 2003-2004. From then on, the survey seems more regular and representative of the observations made in the field by Avalanche Québec technicians.

 # 3 Data and Methods

Four supervised machine learning (ML) methods were used and compared to develop a snow avalanche forecasting tool (event data) using different predictors (meteorological variables): logistic regression (LR), classification trees (CT), random forests (RF) and neural networks (NN) (Figure 3). Our forecasting goal is to predict both dry and wet avalanches in a single and simple daily model, which will predict the probability of an avalanche occurring.

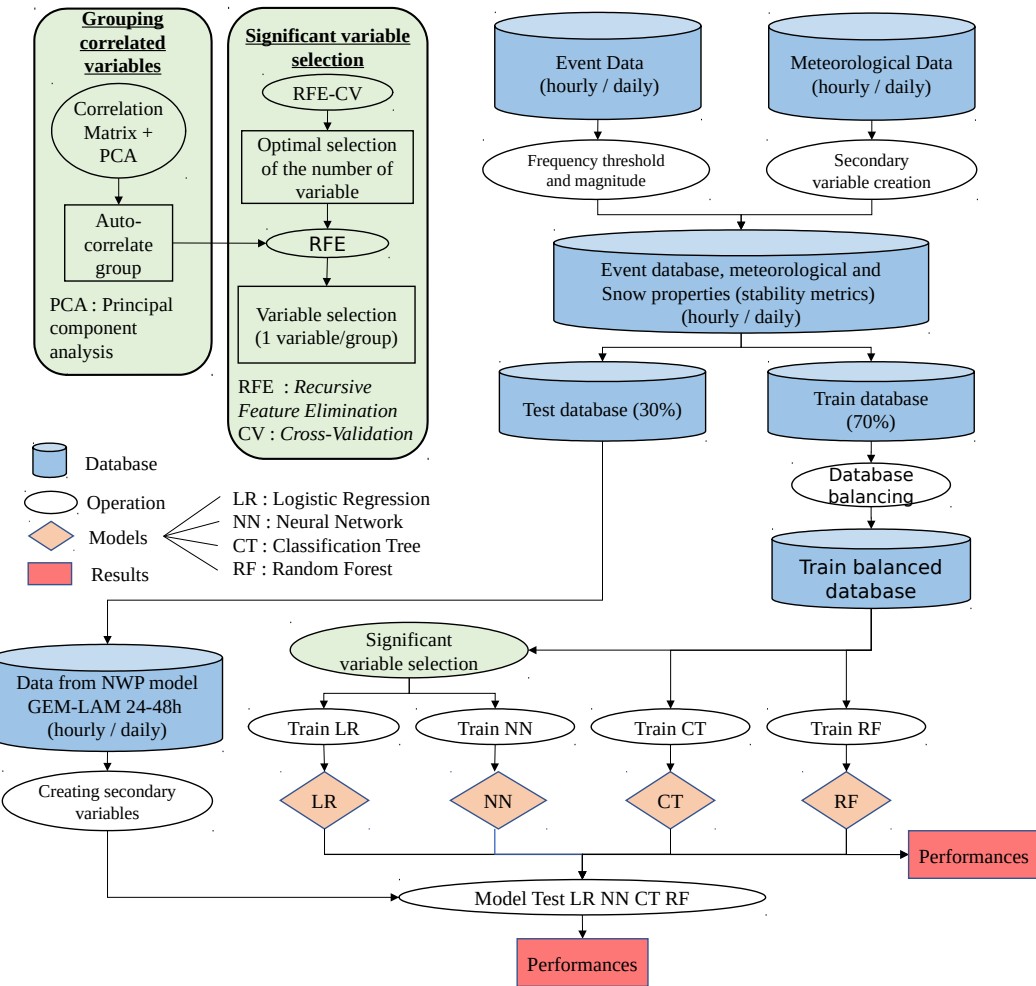

**Figure 3.** Flow chart of the method used to build the machine learning models from avalanche event dataset, meteorological dataset and the dataset from numerical weather prediction NWP (GEMLAM 24-48h.

## 3.1 Avalanche event dataset

The avalanche event database used in this study records all snow avalanche events from the winter 2003-2004 to 2019-2020. The avalanche observations are collected by MTMQ road patrol to assess the road conditions for rock and ice falls, snow avalanches, as well as storm surge at all times. The observations are recorded at the hour of discovery of the debris, within a few hours of the actual avalanche. Even when the road is closed to the public, road patrol still moves along the road to assess road conditions and make observations. Then, the hourly events are summed to a daily (midnight cutoff) dataset that contains the number of snow removal interventions on the road, the avalanche corridor, whether or not the road was reached, and the distance traveled (ditch, shoulder, 1 lane, 2 lanes). The type (dry or wet) and avalanche size are not recorded by the road patrol, and therefore, cannot be used in the analysis. The majority of these observations of avalanches were on the slope and have not reached the road and required intervention. Therefore, for a day to be considered an event, a snow avalanche must have reached the road (shoulder, 1 lane, or 2 lanes). Observations of avalanche deposition in the ditch or on the slopes were not included in the analyses. A binary event day variable ($E_{AVA}$ = 'avalanche day' or 'no avalanche day') was created for days where an avalanche has reached the road, regardless of the size or path length because this information was not recorded (Table 1). If more than one intervention (avalanches) reached the road, duplicate event days were added to the dataset with the following weight of 1, 2, 3, or 4 duplicate days with respect to one intervention, two to five interventions, 6 to 9 interventions, and 10 or more interventions (Table 1). The weights were assigned manually by looking at the frequency distribution of events (Figure 4). Finally, for the *training dataset only*, in order to get a symmetric binomial distribution and avoid over-representation of days where no avalanche was observed, the same number of days were randomly selected where avalanches were not observed on the road. This duplicated days by weight and the balance of event days were only done on the *train* dataset. The constitution *test* is described below in section 3.4.

**Table 1.** Description of the event variable tested.

| Event variable | Conditions |
|---|---|
| **Variable $E_{AVA}$** | – 1 intervention per day $\rightarrow$ weight of 1 <br> – 2 to 5 interventions per day $\rightarrow$ weight of 2 <br> – 6 to 9 interventions per day $\rightarrow$ weight of 3 <br> – 10 interventions per day and + $\rightarrow$ weight of 4 |

## 3.2 Avalanche danger level dataset

To go along with the avalanche event dataset, Avalanche Quebec issued avalanche danger levels along road 132 from 2015 to 2020. The forecasts are issued every day by Avalanche Québec's forecaster teams specifically for the avalanche paths along

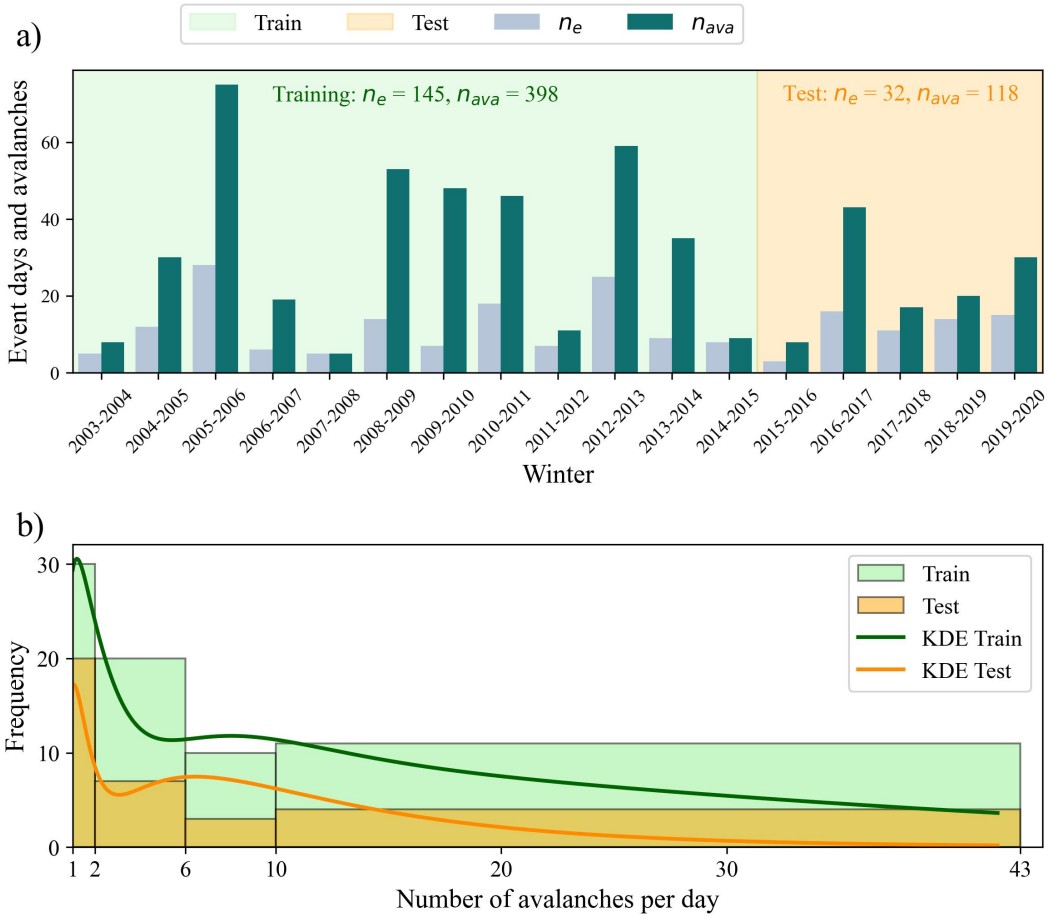

**Figure 4.** a) Time series of event days ($n_e$) and number of avalanches ($n_{ava}$) for every year from 2003 to 2020, with the indication the training and test dataset temporal range. b) Frequency distribution of the number of avalanches per day with a kernel density estimation (KDE), and a histogram with the chosen bin weight for the duplication of event days.

road 132. They based their prediction on a specific danger level scale relative to the occurrence of avalanches on these specific coastal paths along the road 132 (Figure 2). The danger level for the road was adapted from the five-level danger scale of North America (Statham et al., 2010), to four danger levels (level 4 and 5 combined) (Table 2. The decision of combining both level 4 and 5 was a joint decision made by Avalanche Québec's team of forecasters and the local direction of Québec's Ministry of Transport (St-Anne-des-Monts office). From this scale, the cutoff between an avalanche occurrence on the road is at the *Considerable* level where small avalanches are expected on the roads, followed by the *High* level with multiple small or big avalanches on the road. The *Moderate* level is characterized by a possibility of small avalanche activity but not reaching the road, followed by a *Low* danger level where avalanche activity is unlikely. With the forecast danger level of Avalanche Québec, we considered an event predicted at the *Considerable* level (and *High*), where small avalanches are expected to reach the road

(Table 2). Considering an event with the danger level *Considerable* and *High* allows us to compute the same performance metrics as the ML prediction, thus comparing each other's performance. We also consider an event day where the probability of the ML model prediction is above 50%, enabling us to compute the performance metrics (section 3.4.6) and comparing the ML algorithm probability to the Avalanche Quebec forecasted danger level, based on the avalanche event dataset as ground truth.

**Table 2.** Avalanche danger level for transportation corridor in northern Gaspésie and expected avalanches occurrences.

| Danger level | Expected avalanches |
|---|---|
| Low | No avalanches and/or unlikely small avalanches outside the road. |
| Moderate | Possible small avalanches outside the road and/or unlikely big avalanches outside the road. |
| Considerable | Possible small avalanches **on the road** and/or big avalanches possible outside the road. |
| High or Extreme | Multiples small avalanches likely **on the road** and/or big avalanches likely **on the road**. |

### 3.3 Meteorological data

The independent variables provided as input to the ML models were derived from daily (at midnight) meteorological data recorded at the Cap-Madeleine station (20 m a.s.l): air temperature (°C), wind speed (km/h) and direction (°), precipitation (mm) and relative humidity (%). These were used to produce 88 direct, cumulative, and derived meteorological variables (Table 3). The weather station is located to the east of the study areas (Figure 1), at the sea level ( 0 m a.s.l). The altitude of the release areas in average 50 m a.s.l, and goes up to 200 m in Mont-Saint-Pierre sector (Figure 1.

To assess the predictive capability 24h in advance of ML models with NWP, we used the Canadian HRDPS (High Resolution Deterministic Prediction System) based on the Regional Deterministic Prediction System (RDPS) configuration of the 5.0.2 version of the Global Environmental Multiscale model (GEM5) (McTaggart-Cowan et al., 2019; Girard et al., 2014). The expert model trained on the meteorological data from the Cap-Madeleine station was tested using the HRDPS weather forecast data over 24h (GEMLAM 24h) and 48h (GEMLAM 48h). GEMLAM data with a spatial resolution of 2.5 km were acquired for a tile centered on the Cap-Madeleine station (20 m a.s.l.), with a tile altitude of 11 m a.s.l.

Mean (Tmean), minimum (Tmin) and maximum (Tmax) air temperatures were calculated over periods of 24, 48, 72, 96 and 120 hours. Daily mean (Tmean), minimum (Tmin) and maximum (Tmax) temperatures from the previous and up to three previous days were also considered and noted for example: Tmean_-1d. Tmin and Tmax (4h/24, 8h/24, 12h/24, 4h/48, 8h/48, 12h/48) are the minimum and maximum temperatures calculated over moving averages of 4, 8 and 12 hours over periods of 24 and 48 hours. Thawing degree-days are defined as the sum of daily mean temperatures (Tmean_24h) with a minimum limit

value set at 0 °C (Gauthier et al., 2015). The calculation of degree-days begins at the start of the operational season as defined later in the section. Frost intensity (Int_frost) is equal to Tmin_24h when Tmax of the previous day is positive and Tmin of the current day is negative. Thaw intensity (Int_thaw) is equal to Tmax_24h when Tmin of the previous day is negative and Tmax of the current day is positive. Thermal amplitude (DTR_d) is the difference between the daily maximum and minimum temperatures up to three days before the current day.

Rain and snow correspond to precipitation when air temperature is above and below 0°C respectively. They were accumulated over periods of 24, 48, 72, 96 and 120 hours. Hourly rain and snow intensities were calculated for different time blocks: 1h/24, 4h/24, 8h/24, 12h/24, 24h/24, 1h/48, 4h/48, 8h/48, 12h/48, 24h/48, 48h/48, 72h/72, 96h/96, 120h/120. Average (WV_mean) and maximum (WV_gust_max) wind velocity (km/h) and maximum wind gust direction (WV_gust_max) (°) were defined for each 24-hour period. Finally, the SnowDriftIndex is an important variable representing the potential amount of snow transported by the wind (Hendrikx et al., 2014; Pomeroy, 1989). The index is equal to the product of the cube WV_mean multiplied by the accumulated snow. The index was calculated for periods of 24, 48, 72, 96 and 120 hours. The threshold for the solid and liquid precipitation phase was set at 1°C for the calculation of the SnowDriftIndex (Pomeroy, 1989).

The condition used to determine the start of the snow avalanche season is based on the amount of snow required to fill the merlons, which is set at 50 mm of snow water equivalent (SWE), following the study of Gauthier et al. (2017). Since 2003-2004, no avalanches have been observed on the road before this threshold was reached. The only "exception" is an avalanche that terminated on the shoulder after 30 mm SWE in November 2019. The season ends on April 15 with the end of Avalanche Québec's operating season. Only nine avalanches have been observed after this date since 2004. However, none of these avalanches have been observed over the five years of the Test dataset. These limits were used to establish the temporal extent of the dataset used to test the models. To train the models, we used the entire dataset, starting and ending the season with the first and last avalanches recorded on the road.

## 3.4 Learning procedure

The challenge in developing a good predictive model is to strike a balance between selecting a minimum number of non-redundant explanatory variables and maintaining an optimal level of performance. Principal component analysis (PCA) and cross-correlations (correlation matrix) were used to group variables that showed colinearity. Based on the results of these analyses and our understanding of the physics of snow avalanche development mechanisms (e.g. Lehning et al., 1999; McClung and Schaerer, 2006), seven groups of variables were defined (Table 3). Thus, only one variable per correlated group can be used by the different ML models.

The next step is to divide the daily event and weather dataset into training (Train) and validation (Test) datasets (Figure 3). Nguyen et al. (2021) have shown that a ratio of 70/30 gives the best performance. This approach aims to train the models on a portion of the data with the aim of identifying the best meteorological variables capable of explaining the occurrence of snow avalanches, and then test the model in a predictive context (hindcast) (Kotsiantis, 2007). The years used to train the model and those to test it were selected according to the years of avalanche forecasts on roads 132 by Avalanche Québec (AvQ). Thus, the Train dataset covers 10 winters between 2003-2004 and 2014-2015. The models were then tested over AvQ's last five winters

**Table 3.** Meteorological variables with their abbreviation (abbr), followed by their description. The colors represent the different weather variables groups and the different shades within these groups represent variables that are auto-correlated.

| Meteorological variables (abbr) | Description |
|---|---|
| Tmean, Tmin, Tmax (24h, 48h, 72h, 96h, 120h)<br><br>Tmean, Tmin, Tmax (-1d, -2d, -3d)<br><br>Tmin, Tmax (4h/24, 8h/24, 12h/24, 4h/48, 8h/48, 12h/48) | Temperature mean, min and max over 24h, 48h, 72h, 96h, 120h (°C)<br><br>Temperature mean, min and max over 24h minus 1 day, 2 days, 3 days(°C)<br><br>Temperature min and max (°C) for 4h, 8h and 12h over a period of 24h and 48h |
| DD | Degrees-days of thaw (°C) |
| Int_frost, Int_thaw<br><br>DTR (d, d-1, d-2, d-3) | Intensity of frost or thaw between actual day and the day prior (°C)<br><br>daily thermal amplitude and until 3 days (°C) |
| Rain (24h, 48h, 72h, 96h, 120h)<br><br>Int_rain (1h/24, 4h/24, 8h/24, 12h/24, 24h/24, 1h/48, 4h/48, 8h/48, 12h/48, 24h/48, 48h/48, 72h/72, 96h/96, 120h/120) | Rainfall over 24, 48, 72, 96 and 120h (mm)<br><br>Rain intensity (mm/h) for 1h, 4h, 8h and 12h over a period of 24h, 48h, 72h, 96h and 120h |
| Snow (24h,48h)<br><br>Int_snow (1h/24, 4h/24, 8h/24, 12h/24, 24h/24, 1h/48, 4h/48, 8h/48, 12h/48, 24h/48, 48h/48, 72h/72, 96h/96, 120h/120) | Snowfall over 24 and 48h (mm) *not correlated with Snow (72,96,120h)<br><br>Snow intensity (mm/h) for 1h, 4h, 8h and 12h over a period of 24h, 48h, 72h, 96h and 120h |
| Snow (72h, 96h, 120h) | Snowfall over 72, 96 and 120h mm) *not correlated with Snow (24,48h) |
| WV, WV_gust_max (24h) | Wind velocity mean and max over 24h (km/h) |
| SnowDriftindex (24h, 48h, 72h, 96h, 120h) | Snow drift index 24, 48, 72, 96 and 120h |
| WD_gust_max (24h) | Wind direction of the max gust over 24h (°) |

of operation: 2015-2016, 2016-2017, 2017-2018, 2018-2019, 2019-2020. This selection of winters will enable us to compare the model performance with traditional avalanche forecasting.

Then, the training dataset was balanced to contain as many events as non-events. This practice avoids prioritizing one dependent variable over another. To achieve this, we randomly selected the same number of non-event days as event days for each month in the training database. This operation was repeated 50 times for the four proposed ML methods. The performance of the 200 models generated was then tested on the Test dataset to select the best performers.

Finally, some ML algorithms such as LR and NN require explanatory variables to be pre-selected to avoid colinearity between variables. A recursive feature elimination with cross-validation (RFE-CV) method was first used to reduce the total number of variables initially at 96 variables. RFECV iteratively removes the least important features while evaluating model performance using cross-validation at each step. In our case, feature importance was determined using a Random Forest, and the F1-score was used as the evaluation metric to decide when to stop the elimination loop. The optimal number of features was selected as the one yielding the highest average $F_1$ score. This process was applied before training the LR and NN models, enabling the selection of a minimal number of variables while maintaining an optimal level of performance (Dormann et al., 2013). Once RFECV was complete—i.e., when the F1-score no longer improved by removing additional variables—we applied Recursive Feature Elimination (RFE) to retain the feature with the highest importance from each correlated group (Table 3). The final number of features thus depended on how many variables remained after RFECV. Specifically, if at least one variable per group survived RFECV, the final selection included 8 features (one per group). If all variables in a given group were removed during RFECV, the final model contained fewer features. In other words, the variable with the highest importance was kept per correlated group. Finally, the RFE-CV and RFE must be based on an ML method for calculating and comparing the importance of variables (FI for feature importance) in relation to each other (Table A1). It is recommended to use a different ML method from those considered for model development (Midi et al., 2010). RFs were used to establish the importance of variables since they are less affected by the presence of multicolinearity, and this pre-selection step is mainly applied to LR and NN. The RFE was done using the function RFE from the package *scikit-learn* in python (Pedregosa et al., 2011). Finally, some ML algorithms such as LR and NN require explanatory variables to be pre-selected to avoid colinearity between variables. A recursive feature elimination with cross-validation (RFE-CV) method was first used to reduce the total number of variables initially at 96 variables. RFECV iteratively removes the least important features while evaluating model performance using cross-validation at each step. In our case, feature importance was determined using a Random Forest, and the F1-score was used as the evaluation metric to decide when to stop the elimination loop. The optimal number of features was selected as the one yielding the highest average $F_1$ score. This process was applied before training the LR and NN models, enabling the selection of a minimal number of variables while maintaining an optimal level of performance (Dormann et al., 2013). Once RFECV was complete—i.e., when the F1-score no longer improved by removing additional variables—we applied Recursive Feature Elimination (RFE) to retain the feature with the highest importance from each correlated group (Table 3). The final number of features thus depended on how many variables remained after RFECV. Specifically, if at least one variable per group survived RFECV, the final selection included 8 features (one per group). If all variables in a given group were removed during RFECV, the final model contained fewer features. In other words, the variable with the highest importance was kept per correlated group. Finally, the RFE-CV and RFE must be based on an ML method for calculating and comparing the importance of variables (FI for feature importance) in relation to each other (Table A1). It is recommended to use a different ML method from those considered for model development (Midi et al., 2010). RFs were used to establish the importance of variables since they are less affected by the presence of multicolinearity, and this pre-selection step is mainly applied to LR and NN. The RFE was done using the function RFE from the package *scikit-learn* in python (Pedregosa et al., 2011).

### 3.4.1 Expert model

We used "expert model" to test the hypothesis that simpler non-processed meteorological variables could perform well in ML prediction for avalanche events in an operational avalanche management context. In addition, processed meteorological variables may become outliers or present a high degree of uncertainty when used with numerical weather prediction (NWP) as input. However, the numerical weather prediction offers a unique opportunity to use these ML models as operational forecasting tools for risk management along transportation corridors. It is in this context of snow avalanches operational forecasting that expert models are proposed. The variables used in these expert models were selected following two steps : 1) the redundancy of the most important variables as determined by the four ML algorithms (Table A1), and 2) our understanding from previous work on the development of instabilities in the snowpack for both dry and wet avalanches in the study area (i.e. Gauthier et al., 2017), ultimately leading us to snow/rain accumulation (loading), and temperature (melting) (Ancey, 2006; McClung and Schaerer, 2006). The expert models were trained and then tested on the *Train* and *Test* datasets with meteorological data from the Cap-Madeleine station. They were then tested against the avalanche forecasts of Avalanche Québec over the winter 2018-2019 with weather data from the Cap-Madeleine station (Hindcast).

### 3.4.2 Logistic regression (LR)

LR relies on a logistic function to calculate the probability of occurrence of an event (Hosmer and Lemeshow, 2000). It is a statistic that does not tolerate multicolinearity well, and requires a reduction in the number of variables provided as input to achieve good performance (Midi et al., 2010). A scaling of meteorological variables was also carried out when training the LRs to avoid problems of over- or under-representation of the latter in the models (Menard, 2011). We used the logistic regression function from the scikit-learn package in python (Pedregosa et al., 2011), with all default hyperparameters.

### 3.4.3 Classification Trees (CT)

CTs take the form of a tree whose nodes refer to a logical function or decision threshold (branch). They can be easily used as decision diagrams, hence their reputation as a simple and effective tool for operational hazard management (e.g. Hendrikx et al., 2014). Various classification algorithms can be used to divide the nodes. In our case, the Gini index represents the function best suited to dividing binary variables (e.g. Hendrikx et al., 2014). Unlike other ML algorithms, CTs are not affected by the use of outliers or collinear variables (Mendeş and Akkartal, 2009), allowing them to manage the selection of significant variables themselves. Overlearning on training data is a frequent problem with this algorithm. The best way to counter it is to limit tree growth with pre-pruning methods (Kotsiantis, 2007). We have limited tree growth to three levels to maintain optimal performance with a limited number of predictors. We used the Decision Tree Classifier function from the scikit-learn package in python (Pedregosa et al., 2011), with all default hyperparameters, except max_leaf_nodes was set to 5 to limit the model's complexity and reduce overfitting.

### 3.4.4 Random forests (RF)

RFs are composed of a set of classification (decision) trees. Each tree is trained with different groups of randomly established variables (bootstrap), enabling the use of collinear variables (Ma et al., 2021; Revuelto et al., 2020; Strobl et al., 2008). For each prediction, each tree in the forest decides on a class, and the one predicted by the majority of all trees wins. The use of a large number of trees (ntree) generally improves model performance, but also significantly increases processing time (Hasan et al., 2016; Yoo et al., 2012). Hasan et al. (2016) and Couronné et al. (2018) mention that a few hundred trees (e.g.: 500) are usually sufficient and that a larger tree only increases processing time. They are fast and easy to use, with good tolerance of outliers and noise in the data (Breiman, 2001). On the other hand, overlearning on training data is also a frequent problem with this algorithm. We used the Random Forest Classifier function from the scikit-learn package in python (Pedregosa et al., 2011), all default hyperparameters were used except for max_depth=5 to control the complexity of each tree, and n_estimators=400 to improve model stability and performance by averaging across more trees.

### 3.4.5 Neural network (NN)

NNs are composed of several layers of neurons in which data are processed. The type of NN used here is the multilayer perceptron (MLP) with a non-linear activation function similar to a logistic function. Like LR, NNs have the advantage of non-linear variable preselection (Kavzoglu and Mather, 2002) and scaling of meteorological variables. We used the MLP Classifier function from the scikit-learn package in python (Pedregosa et al., 2011). The architecture consisted of a single hidden layer with 500 neurons (hidden_layer_sizes=500), using the hyperbolic tangent function (activation='tanh') as the activation function. All other hyperparameters were left at their default values.

### 3.4.6 Performance indicators

The various metrics used to assess model performance are based on the count of events and non-events correctly predicted or not, as defined in the confusion matrix (Table 4). The probability of non-event detection (PON) is a relevant indicator for assessing the model's performance in predicting non-event days, which generally represent the majority of days in a year. However, the forecasting objective is to find the best compromise between the number of detections (motorist safety) and false alarms (additional operating costs). Thus maximizing the number of events predicted (Precision or True Positive Rate TPR) while minimizing false alarms (False Alarm Rate FAR or False Positive Rate FPR). Improvement in one often leads to deterioration in the other. The Receiver Operating Characteristic (ROC) and the $F_1$ make it possible to establish a relationship between predicted events (Prec or TPR) and false alarms (FPR or FAR). The ROC is a graphical representation that relates TPR and FPR, but the area under the curve (AUC) of this relationship (ROC) incorporates a single value that facilitates the comparison of overall model performance. The work of Saito and Rehmsmeier (2015) informs us that the use of AUC-ROC as a performance indicator should be employed with caution on unbalanced test datasets. In the present case, the dataset used to test the models and select the best model generated after the 50 iterations contains a much greater number of non-events than events. In this case, the AUC of the ROC will be biased by the high value of predicted non-events included in the RPF

275  calculation. The $F_1$ is the relationship between PREC and TPR (Table 5). $F_1$ is the most widely used indicator for finding the model allowing the best compromise between these two metrics for unbalanced datasets (He and Ma, 2013). When $F_1$ was equal for two models, we suggest prioritizing the one with the best PREC to limit false alarms.

**Table 4.** Confusion Matrix with True negative TN, False negative FN, False alarm FP and True positive TP.

| | | Observations | |
|---|---|---|---|
| | | **Non-occurred** | **Occurred** |
| **Predictions** | **Non-occurred** | TN: Predicted non-event | FN: Non-predicted event |
| | **Occurred** | FP: False alarm | TP: Predicted event |

**Table 5.** Performance metrics with their respective formula from the confusion matrix.

| Performance metrics | Formula |
|---|---|
| **Precision (PREC)** | $\frac{TP}{(TP+FP)}$ |
| **True positive rate (TPR)** | $\frac{TP}{(FN+TP)}$ |
| **Probability non-event (PON)** | $\frac{TN}{(TN+FP)}$ |
| **False positive rate (FPR)** | $\frac{FP}{(FP+TN)}$ |
| **False alarm rate (FAR)** | $\frac{FP}{(FP+TP)}$ |
| **$F_1$** | $\frac{2\cdot(PREC\cdot TPR)}{(PREC+TPR)}$ |
| **ROC(AUC)** | TPR vs FPR |

## 4 Results

Since the aim is to compare the performance of the four ML methods, expert model, and the predictive capability using high
resolution NWP, we first present the performance indicators and selected variables for the training (Train) and the validation (Test) dataset for each of the four ML methods. The variables used by the RFs are not shown, since the model uses all the predictor variables. In the subsequent section 4.2, we present the performance indicators for the expert models using the entire dataset,and a hindcast for the winter 2018-2019 using weather station and numerical weather prediction for 24 and 48h in advance (GEMLAM 24-48).

## 4.1 Training models

The best model was selected by the highest $F_1$ score and the corresponding highest score of ROC(AUC) if needed. For the hindcast of snow avalanches on road 132, CT is the best algorithm, followed very closely by NN, with $F_1$ values of 0.41 and 0.40 respectively. With 28 well-predicted events out of 32 (TPR of 0.88), NN has a better event detection capability than CT, which predicted 22 (TPR of 0.69). The NN prioritizes event detection over false alarms (FPR of 0.18), while the opposite logic applies to the CT, which has a lower false alarm rate (FPR of 0.12).

**Table 6.** Results and performance indicators for the model selected with recursive feature selection (RFE). In bold, the best $F_1$ score and ROC(AUC) for the *Test* dataset. The performance metrics are computed with the avalanche event dataset where avalanches reached the road. An event day predicted by the model is considered when the probability of the model is above 50%. The confusion matrix is presented under C. matrix. The performance indicators abbreviation are defined in Table 5. The weather variables abbreviation are defined in Table 3.

| ML | Dataset | C. matrix | | Performance indicators | | | | | | | Variables |
|---|---|---|---|---|---|---|---|---|---|---|---|
| | | TN | FN | PREC | TPR | PON | FPR | FAR | $F_1$ | ROC(AUC) | |
| | | FP | TP | | | | | | | | |
| LR | Train | 48 | 18 | 0.89 | 0.88 | 0.75 | 0.25 | 0.11 | 0.88 | 0.81 | Int_snow_8h/24h |
| | *Balance (10 yrs)* | 16 | 127 | | | | | | | | Tmax_12h/24h |
| | **Test** | 356 | 4 | 0.24 | 0.88 | 0.80 | 0.20 | 0.76 | 0.37 | 0.84 | Snowdrift_24h |
| | *Hindcast (5 yrs)* | 91 | 28 | | | | | | | | Snow_72h |
| CT | Train | 58 | 34 | 0.95 | 0.77 | 0.91 | 0.09 | 0.05 | 0.85 | 0.84 | Snowdrift_24h |
| | *Balance (10 yrs)* | 6 | 111 | | | | | | | | Tmean_48h |
| | **Test** | 393 | 10 | 0.29 | 0.69 | 0.88 | 0.12 | 0.71 | **0.41** | 0.78 | Tmean_72h |
| | *Hindcast (5 yrs)* | 54 | 22 | | | | | | | | |
| RF | Train | 63 | 6 | 0.99 | 0.96 | 0.98 | 0.02 | 0.01 | 0.98 | 0.97 | Int_snow_12h/24h |
| | *Balance (10 yrs)* | 1 | 139 | | | | | | | | Int_snow_8h/24h |
| | **Test** | 377 | 7 | 0.26 | 0.78 | 0.84 | 0.16 | 0.74 | 0.39 | 0.81 | Snow_120h |
| | *Hindcast (5 yrs)* | 70 | 25 | | | | | | | | [...] |
| NN | Train | 49 | 20 | 0.89 | 0.86 | 0.77 | 0.23 | 0.11 | 0.88 | 0.81 | Int_snow_8h/24h |
| | *Balance (10 yrs)* | 15 | 125 | | | | | | | | Snow_72h |
| | **Test** | 366 | 4 | 0.26 | 0.88 | 0.82 | 0.18 | 0.74 | 0.40 | **0.85** | Tmax_12h/24h |
| | *Hindcast (5 yrs)* | 81 | 28 | | | | | | | | Snowdrift_24h |

Maximum snow intensities 12/24 or 8/24 hours (Int_snow) are the most redundant and important variables selected by three of the four models: LR, RF and NN (Table 6-A1). Total snowfall over 72 or 120 hours (Snow) and windy snow indices over 24 and 72 hours (SnowDrift) are frequently recurring variables with a very high level of importance for all models. Average temperatures over 48 and 72 hours were used by CT (Table 6). These are not, however, variables that were frequently selected or had a high level of importance when the models were trained (Table 6).

## 4.2 Expert models

The four variables were selected(Snow_24h, Snow_72h, Tmean_48h and Rain_24h) to feed the development of an expert model. This also facilitates the comparison of ML methods with each other to demonstrate their predictive capacity, with results comparable to those obtained by the more complex models developed to explain the occurrence of snow avalanches (Table 6). The $F_1$ score and the ROC(AUC) of the expert model were similar to the other models in Table 6. With this variable preselection, LR and RF show the best performance metrics with $F_1$ and ROC score. The best method was the RF with $F_1$ of 0.41 with the *Test* dataset, compared to 0.39 for the other three methods, but among them, LR had the highest ROC(AUC) with 0.85 (Table 7). Interestingly, the expert model (RF) had the same $F_1$ score of 0.41 as the other "unrestricted" model (CT) in Table 6, but the expert model with RF had the highest ROC(AUC) of 0.85 (Table 7). Interestingly, Tmean_48h was left out by the CT (Table 7), yet this was the only ML method to have selected the variable in the "unrestricted" model to explain avalanche occurrence (Table 6). The expert model performance was also assessed for the specific hindcast winter of 2018-2019, as a basis for comparison with NWP data. The best method was the RF method, which was the same as for the *Train* dataset, with a $F_1$ score of 0.47 and a ROC(AUC) of 0.88 (Table 7). The CT method was second with a $F_1$ score of 0.45 and a ROC(AUC) of 0.83.

To assess the probability prediction with a human-based forecast, we compared them with Avalanche Québec forecasts for the study area. Their forecast performance is in Table 8, where the performance indicators are comparable to the ML methods using the *Train* dataset. The AvQc $F_1$ score for the *Train* dataset (5 hindcast winter) was 0.42, compared to the best one with the expert model (Rf with 0.41) and the "unrestricted" model (CT with 0.41). Figure 5 shows the seasonal evolution of the probability from the expert models using the 4 ML methods, with also the danger level forecasted by Avalanche Québec transferred into probability. At first glance, the probabilities of the ML methods and Avalanche Québec had a great fit visually, with the CT having the greatest fit visually. Both the ML methods and Avalanche Québec predicted high probability around January 10th with 23 mm of snow in 24h, but no avalanche was recorded. This systematic false alarm was estimated by the 4 ML methods and Avalanche Québec. The remaining avalanche days were either estimated by some ML methods or Avalanche Québec forecast.

## 4.3 Forecast performance with NWP

The models presented here have been developed based entirely on an analysis of the statistical performance of the models and their a posteriori predictive capability. In reality, the goal is to use a simpler expert model with numerical weather prediction (NWP) in a forecast context. Errors (deviations) of NWP's temperature data are very low with GEMLAM 24 and 48h, but much more uncertainty is present with precipitation data, especially regarding precipitation intensity (mm/h) (Gauthier et al., 2022). By using an expert model based solely on simple weather variables such as Snow_24h (accumulation in mm), while avoiding more rate intensity variables like Int_snow (mm/h) to limit potential forecast errors.

For the winter 2019, the performance of the models with GEMLAM 24 and 48 h forecast data is significantly lower than that established with meteorological data (hindcast), with around 0.4 for the Hindcast 2019 and around 0.3 for GEMLAM(24-48h)

**Table 7.** Performance of **expert models** on the event variable ($E_{AVA}$) for the ***Train* (balance 10 years) and *Test* dataset (5 years)**. In bold, the best F$_1$ score and ROC(AUC) of the *Test* dataset. The performance metrics are computed with the avalanche event dataset where avalanches reached the road. An event day is considered when the probability of the model is above 50%. The confusion matrix is presented under C. matrix. The performance indicators abbreviation are defined in Table 5. The Variables are defined in Table 3. * Rain_24h was not selected by the CT algorithm.

| ML | Dataset | C. Matrix | | PREC | TPR | PON | FPR | FAR | F$_1$ | ROC | Variables |
|---|---|---|---|---|---|---|---|---|---|---|---|
| | | TN | FN | | | | | | | | |
| | | FP | TP | | | | | | | | |
| LR | Train | 41 | 18 | 0.85 | 0.88 | 0.64 | 0.36 | 0.15 | 0.86 | 0.76 | Snow_24h |
| | *Balance (10yrs)* | 23 | 127 | | | | | | | | Snow_72h |
| | **Test** | 364 | 4 | 0.25 | 0.88 | 0.81 | 0.19 | 0.75 | 0.39 | **0.85** | Tmean_48h |
| | *Hindcast (5yrs)* | 83 | 28 | | | | | | | | Rain_24h |
| CT | Train | 55 | 25 | 0.93 | 0.83 | 0.86 | 0.14 | 0.07 | 0.88 | 0.84 | Snow_24h |
| | *Balance (10yrs)* | 9 | 120 | | | | | | | | Snow_72h |
| | **Test** | 380 | 8 | 0.26 | 0.75 | 0.85 | 0.15 | 0.74 | 0.39 | 0.80 | Tmean_48h |
| | *Hindcast (5yrs)* | 67 | 24 | | | | | | | | * |
| RF | Train | 61 | 8 | 0.98 | 0.94 | 0.95 | 0.05 | 0.02 | 0.96 | 0.95 | Snow_24h |
| | *Balance (10yrs)* | 3 | 137 | | | | | | | | Snow_72h |
| | **Test** | 375 | 5 | 0.27 | 0.84 | 0.84 | 0.16 | 0.73 | **0.41** | 0.84 | Tmean_48h |
| | *Hindcast (5yrs)* | 72 | 27 | | | | | | | | Rain_24h |
| NN | Train | 50 | 29 | 0.89 | 0.80 | 0.78 | 0.22 | 0.11 | 0.84 | 0.79 | Snow_24h |
| | *Balance (10yrs)* | 14 | 116 | | | | | | | | Snow_72h |
| | **Test** | 377 | 7 | 0.26 | 0.78 | 0.84 | 0.16 | 0.74 | 0.39 | 0.81 | Tmean_48h |
| | *Hindcast (5yrs)* | 70 | 25 | | | | | | | | Rain_24h |

**Table 8.** Avalanche Québec's forecast performance between 2013 and 2020 (5 operational seasons). The performance metrics are computed with the avalanche event dataset where avalanches reached the road. An event day is considered when the danger level is *Considerable* or *High* (avalanche expected on the road), according to Table 2. The confusion matrix is presented under C. Matrix. The performance indicators abbreviation are defined in Table 5.

| Dataset | C. Matrix | | PREC | TPR | PON | FPR | FAR | F$_1$ |
|---|---|---|---|---|---|---|---|---|
| | TN | FN | | | | | | |
| | FP | TP | | | | | | |
| **Test** | 459 | 65 | **0.79** | 0.29 | 0.98 | 0.02 | 0.21 | **0.42** |
| *Hindcast (5yrs)* | 7 | 26 | | | | | | |

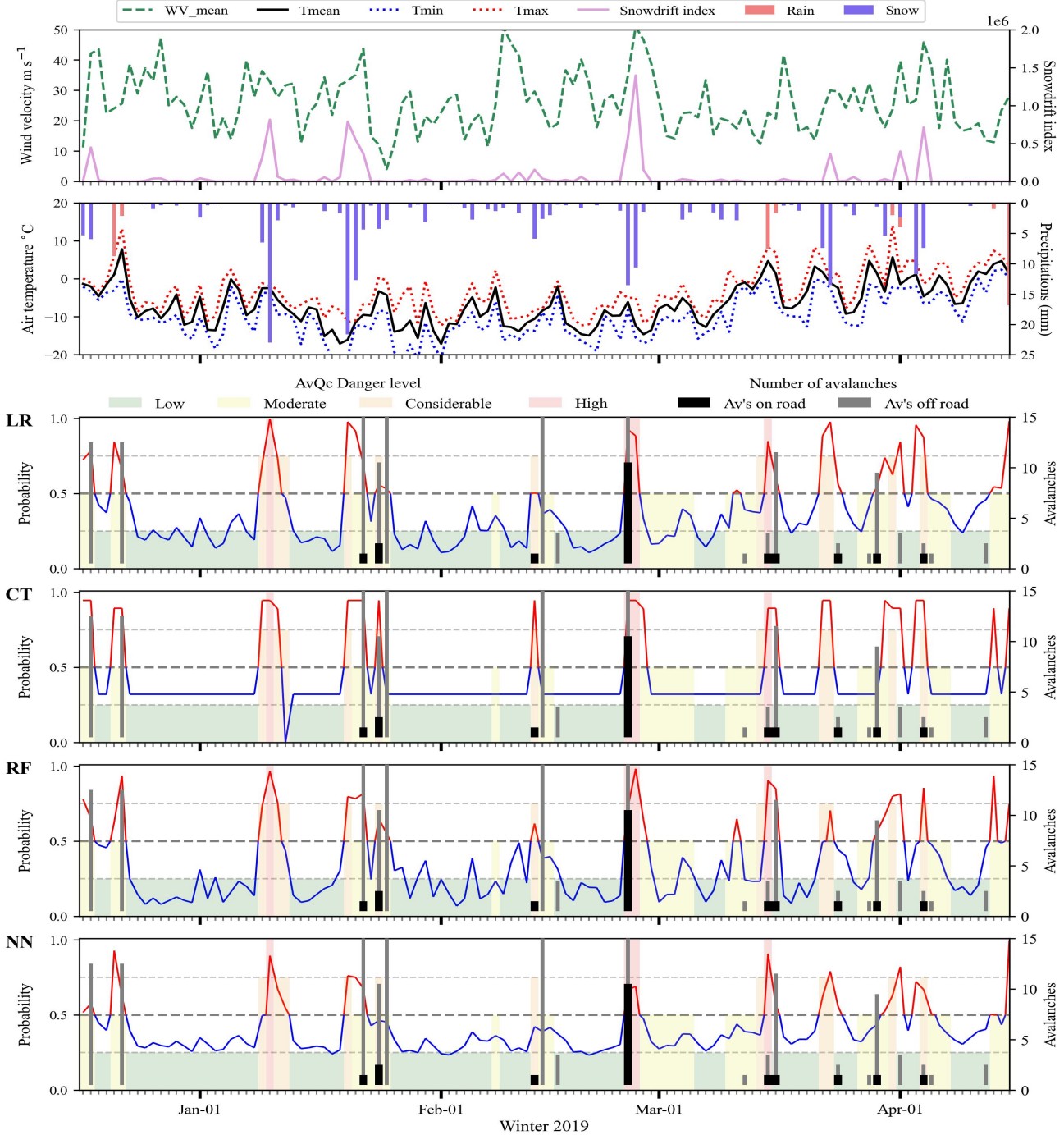

**Figure 5.** Hindcast of snow avalanches on road 132 in winter 2018-2019. Probability calculated with expert models LR, CT, RF and NN. Non-event forecast in blue and event forecast in red. Forecast issued by Avalanche Québec: low (green), moderate (yellow), considerable (orange) and high (red). Event days (occurrence of one or more avalanches on the road) are represented by black histograms, and snow avalanches that did not reach the road by grey histograms.

(Table 9). Surprisingly, the $F_1$ score was higher for the GEMLAM 48h than the 24h, for all 4 ML methods. The best method was LR with a $F_1$ score of 0.31 for GEMLAM 24h and 0.34 for GEMLAM 48h (Table 9).

The probability predicted by the 4 ML methods for the hindcast of 2019 was then compared to the ones predicted by the GEMLAM 24h and 48h (Figure 6). Coefficients of determination $R^2$ greater than 0.8 were obtained with the LR expert model for both GEMLAM 24h and 48h. The NN performed slightly less well with 0.8 for GEMLAM 24h and 0.74 for GEMLAM 48h. The CT and RF had the lowest $R^2$ coefficients with respectively 0.58 and 0.71 for GEMLAM 24h, and respectively 0.56 and 0.64 for GEMLAM 48h.

**Table 9.** Performance of **expert models** on the event variable ($E_{AVA}$) **for Hindcast 2019 vs NWP 2019 (GEMLAM24-48h)**. In bold, the best $F_1$ score and ROC(AUC) of the hindcast and GEMLAM24-48h. The performance metrics are computed with the avalanche event dataset where avalanches reached the road. An event day is considered when the probability of the model is above 50%. The confusion matrix is presented under C. matrix. The performance indicators abbreviation are defined in Table 5. The Variables are defined in Table 3. * Rain_24h was not selected by the CT algorithm.

| ML | Dataset | C. Matrix | | Performance indicators | | | | | | | | Variables |
|---|---|---|---|---|---|---|---|---|---|---|---|---|
| | | TN | FN | PREC | TPR | PON | FPR | FAR | $F_1$ | ROC | |
| | | FP | TP | | | | | | | | | |
| LR | Hindcast 2019 | 108 | 2 | 0.25 | 0.78 | 0.84 | 0.16 | 0.75 | 0.38 | 0.81 | Snow_24h Snow_72h Tmean_48h Rain_24h |
| | | 21 | 7 | | | | | | | | |
| | GEMLAM 24h | 105 | 3 | 0.20 | 0.67 | 0.81 | 0.19 | 0.80 | **0.31** | **0.74** | |
| | | 24 | 6 | | | | | | | | |
| | GEMLAM 48h | 99 | 1 | 0.21 | 0.89 | 0.77 | 0.23 | 0.79 | **0.34** | **0.83** | |
| | | 30 | 8 | | | | | | | | |
| CT | Hindcast 2019 | 114 | 2 | 0.32 | 0.78 | 0.88 | 0.12 | 0.68 | 0.45 | 0.83 | Snow_24h Snow_72h Tmean_48h * |
| | | 15 | 7 | | | | | | | | |
| | GEMLAM 24h | 108 | 4 | 0.19 | 0.56 | 0.84 | 0.16 | 0.81 | 0.29 | 0.70 | |
| | | 21 | 5 | | | | | | | | |
| | GEMLAM 48h | 110 | 4 | 0.21 | 0.56 | 0.85 | 0.15 | 0.79 | 0.30 | 0.70 | |
| | | 19 | 5 | | | | | | | | |
| RF | Hindcast 2019 | 112 | 1 | 0.32 | 0.89 | 0.87 | 0.13 | 0.68 | **0.47** | **0.88** | Snow_24h Snow_72h Tmean_48h Rain_24h |
| | | 17 | 8 | | | | | | | | |
| | GEMLAM 24h | 108 | 4 | 0.19 | 0.56 | 0.84 | 0.16 | 0.81 | 0.29 | 0.70 | |
| | | 21 | 5 | | | | | | | | |
| | GEMLAM 48h | 104 | 3 | 0.19 | 0.67 | 0.81 | 0.19 | 0.81 | 0.30 | 0.74 | |
| | | 25 | 6 | | | | | | | | |
| NN | Hindcast 2019 | 112 | 2 | 0.29 | 0.78 | 0.87 | 0.13 | 0.71 | 0.42 | 0.82 | Snow_24h Snow_72h Tmean_48h Rain_24h |
| | | 17 | 7 | | | | | | | | |
| | GEMLAM 24h | 107 | 4 | 0.19 | 0.56 | 0.83 | 0.17 | 0.81 | 0.28 | 0.69 | |
| | | 22 | 5 | | | | | | | | |
| | GEMLAM 48h | 108 | 3 | 0.22 | 0.67 | 0.84 | 0.16 | 0.78 | 0.33 | 0.75 | |
| | | 21 | 6 | | | | | | | | |

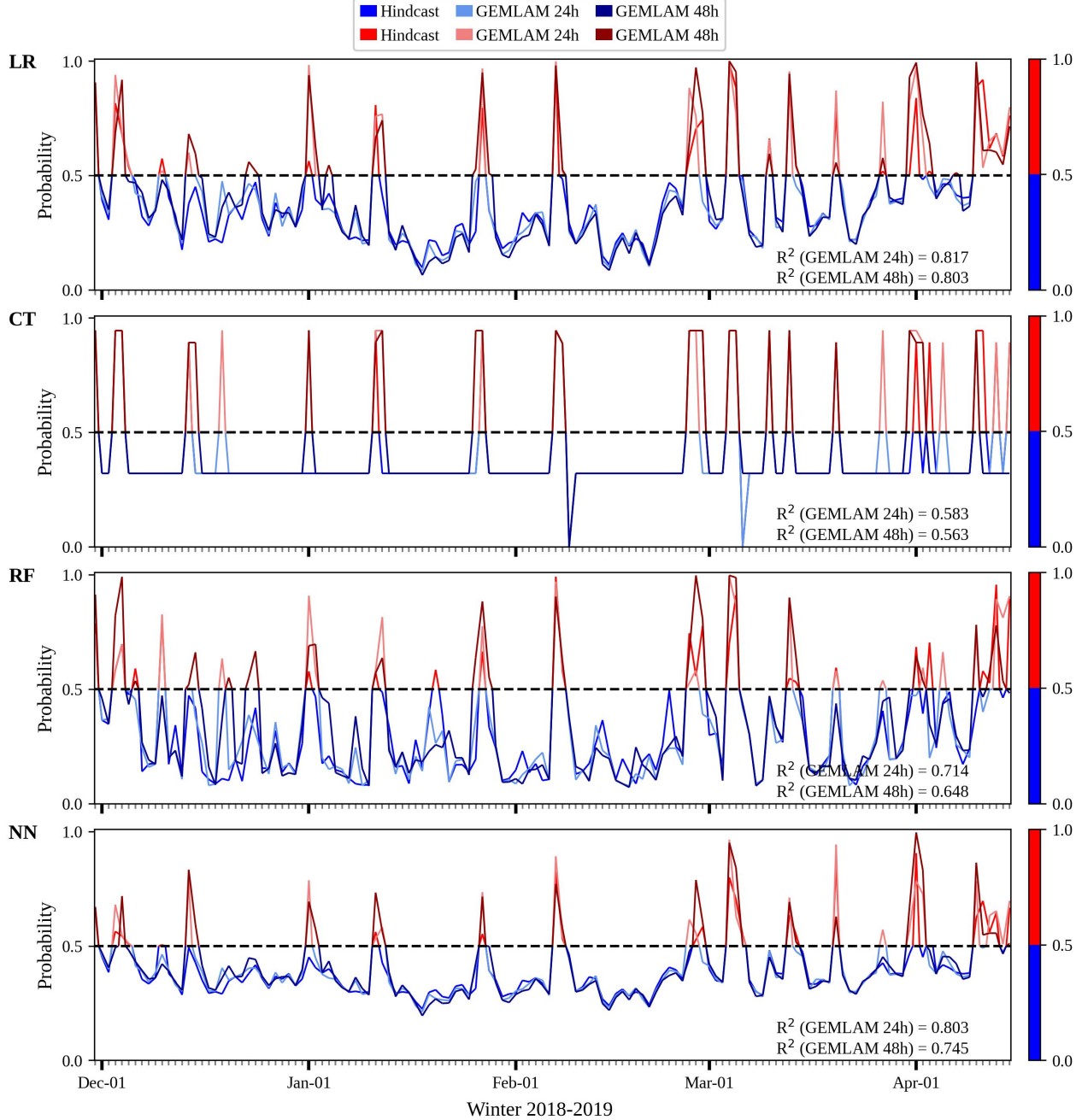

**Figure 6.** Hindcast and HRDPS GEMLAM 24h and 48h forecast of snow avalanches on road 132 in winter 2018-2019. Probability calculated with expert models LR, CT, RF and NN trained on variable $E_{AVA}$. Non-event forecast in blue and event forecast in red.

## 5 Discussion

### 5.1 Variable selection and significance

According to the results obtained, the occurrence of event days is well explained and predicted by the intensity of solid precipitation and snow accumulation over periods ranging from 24 to 120 hours. These results are in line with those obtained by Gauthier et al. (2017, 2018) in the same study area, and with those of many others carried out in the European Alps (Ancey, 2006; Jomelli et al., 2007), in the North American Cordillera (Butler, 1986; McClung and Tweedy, 1993; Hendrikx et al., 2014), in Norway (Bakkehoi, 1987), in New Zealand (Hendrikx et al., 2005) or in Scotland (Ward, 1984). Our results also show a strong tendency towards a selection of variables representative of short-duration, high-intensity precipitation. These variables support one of the two scenarios proposed to explain the snow avalanche occurrences in the northern Gaspé Peninsula (Hétu, 2010; Fortin et al., 2011; Gauthier et al., 2017; Meloche, 2019): 1) a winter regime linked to snowstorms, and 2) a spring regime linked to rain-on-snow events or positive air temperatures. Thus, two types of avalanche problems are likely to explain natural avalanche activity in northern Gaspésie: storm snow (or new snow) avalanches and wet snow avalanches.

Wind snow indices (SnowDriftIndex) have been developed to express windy snow accumulations during storms (Hendrikx et al., 2014; Pomeroy, 1989). Comparing the amplitude of index variations with wind speed and precipitation data, it is clear that SnowDriftIndex is highly reactive during storms with strong winds (Figure 5). But as soon as the precipitation fades, the index rapidly decreases. The SnowDriftIndex represents an important explanatory variable for explaining and predicting the occurrence of storm snow avalanches. This variable could be added to the expert model, but it should not replace the snow accumulation variables (Snow_24h and Snow_72h) to avoid omitting avalanches that occur during snow flurries with no particularly strong winds.

Meloche (2019) has also shown that the strong northwesterly winds coming from the St.Lawrence Gulf, that follow the snowstorms, have the effect of clearing the slopes of snow. Only snow patches protected from the wind by the presence of tree edges between talus slopes persist between storms (Meloche, 2019). Thus, the occurrence of persistent slab avalanches is unlikely on the slopes along road 132. The northwesterly winds rushing into the valley are subject to a Venturi effect that favors wind-driven deflation of the snow on the slope. SnowDriftIndexes have not been developed to support this type of process. Wind speed remains a better indicator of wind deflation after storms. However, wind speed has not proven to be a significant variable frequently selected by ML methods (Table 6).

Finally, rain and air temperatures are variables that can be used to explain and predict wet snow avalanches, both slab and loose snow (Gauthier et al., 2017, 2018). These variables were selected by the best-performing CT (Table 6). Despite their relatively low importance and limited use during the first phase of model development (Table 6), we felt justified in adding Rain and Tmean_48h to the expert model to avoid omitting this type of avalanche problem. Finally, this choice does not seem to have had a significant negative impact on the performance of the models (Table 6-7), and they enable us to predict this type of avalanche. At present, wet snow avalanches are infrequent, but in the context of climate change, it is likely that an increase in the frequency of wet snow avalanches will be observed (Eckert et al., 2024; Locat et al., 2022).

## 5.2 Model Performance

The CT was the best model on the test dataset with only selecting Tmean and Snowdrift variable (Table 6). On the other hand, the other models (LR, RF and NN) have the advantage of returning a better-defined probability of occurrence (Figure 5). For this type of predictive model to be used in an operational management context, the models need to be simple and understandable for managers. With this idea in mind, we developed expert models with four simple variables (Snow_24h, Snow_72h, Tmean_48h and Rain_24h), that proved to be almost as effective as the more complex models generated by the algorithms. With these simpler expert models, the RF algorithms outperformed the other algorithms, both for the test and Hindcast 2019 dataset (Table 7-9). LR had the best performance with the expert model using GEMLAM 24-48h NWP data, but with lower $F_1$ score compared to the train and Hindcast dataset. Figure 6 demonstrates that the GEMLAM model probability was not reaching the maximum of the Hindcast model, thus underestimating some avalanche events and danger levels. This difference could be caused by a well-known bias in NWP where precipitation is significantly underestimated, especially in the study area (Côté et al., 2017). This result demonstrates the ability of simpler models with expert assigned weather variables to predict the avalanche probability in an operational and predictive context. In practice, the variables chosen can be easily modified and the models tested to meet the requirements of the managers. For example, we could force the addition of a variable deemed relevant in the models, such as the snowdrift index or snow intensity (SnowDriftIndex, Int_snow). However, adding a variable and making the model more complex increases uncertainty when the probability of occurrence is calculated using forecast data.

Finally, it is sometimes advisable to avoid using a model that performs too well on training data sets, where model overfitting leads to poorer performance on validation and forecast data. This is often the case with RFs which excel at explaining event days, but whose performance is uncertain when it comes to predicting events. Despite this tendency to overfit (Table 6), RFs represent the best compromise for predicting snow avalanches and limiting false alarms with expert model variables (Table 7). On the other hand, LR is also a wise choice, as this ML method performs best with weather forecast data (Table 7).

## 5.3 Bias and uncertainty

Despite the use of robust cross-validation and feature selection techniques, several sources of bias and uncertainty remain in the modeling framework. First, although a duplication strategy was employed to assign more weight on days with multiple avalanches, this can introduce overfitting in models—particularly in ML algorithms sensitive to redundant information such as neural networks. When duplicated events are repeatedly seen during training, models may learn to overly associate specific patterns with positive labels, reducing their generalization capability to unseen data. This artificial balancing may inflate performance metrics (e.g., $F_1$ score).

Second, the spatial representativity of the meteorological inputs, particularly from the Cap-Madeleine weather station, is a major source of uncertainty. In the western sector of the study area, the nearest AWS is located up to 60 km away from certain avalanche paths, which may result in poor representation of local snow and weather conditions. This introduces potential bias

in both the input data and the interpretation of model performance. Snow and meteorological processes such as precipitation phase and accumulation can vary considerably across such distances, especially in complex alpine terrain.

Additionally, the use of averaged performance metrics across 50 iterations may mask underlying variability in model behavior due to changing train/test splits. While this repetition helps assess robustness, it does not fully capture the impact of outlier events or atypical seasons, which are critical in operational avalanche forecasting. Taken together, these limitations highlight the importance of contextualizing model performance with respect to data representativeness, spatial coverage, and overfitting risks, particularly when applying machine learning to avalanche forecasting.

## 5.4 Using ML prediction for operational avalanche risk management

In an operational management context, the selection of a ML model predicting with accuracy avalanche occurrences is of high interest. Three ML algorithms (LR, RF and NN) returned a better-defined probability of occurrence (Figure 5), with the best scoring performance using 24-48h NWP data (Table 9). They can therefore be used in predictive mode (event or non-event day), but the probability returned by the models could also be useful in an operational context if the probability issued is assigned to an avalanche danger level (Gauthier et al., 2017). Pérez-Guillén et al. (2022) develop an ML model that predicts directly the avalanche danger level, but requires the avalanche danger level in the training dataset. In the context of hazard forecasting and operational management, it is relevant and effective to establish intervention procedures according to different avalanche danger levels, but further investigation is required to map ML probabilities to a danger level.

Danger levels are generally issued by forecasters on the basis of an analysis of snowpack structure and stability, and an analysis of weather forecasts. Using the avalanche forecasts issued by Avalanche Québec for Road 132 between 2015 and 2020, we calculated the performance of their forecasts (Considerable or High) with days with avalanches on the road (Table 2). It appears that the performance of ML's models is nearly the same as the forecasts issued by the organization (Table 8), with similar $F_1$ around 0.4 (hindcast 2015-2020), with RF scoring a 0.40 (best out of 4 ML) and AvQ with a 0.41. Based on this $F_1$ score over 5 winter seasons *Test* dataset, it appears that ML probability is as good as traditional forecasting by AvQ in the area. Similar results were obtained where data-driven assessment was correlated with traditional human-based assessment in several studies in Switzerland (Techel et al., 2022, 2024; Pérez-Guillén et al., 2022; Mayer et al., 2023; Pérez-Guillén et al., 2025). However, the major difference between these studies and this work is that this work does not include snow-cover simulation variables as input, but only weather input variables in the ML model. The addition of snow cover simulation input variables (i.e. SNOWPACK) should be considered in future studies. Using both data-driven and human-based assessment could improve confidence in forecasters' decisions and road management teams (Pérez-Guillén et al., 2025). More and more avalanche forecasting organizations (i.e. France and Switzerland) are using statistical models and snow cover simulations to support their decision-making processes (Hendrick et al., 2023; Mayer et al., 2023; Pérez-Guillén et al., 2022; Viallon-Galinier et al., 2023).

Furthermore, the use of numerical weather prediction (NWP) data rather than station observations or in-situ snowpack profiles makes this approach highly scalable, especially for operations covering large or remote territories (Herla et al., 2024). However, while the model shows better performance in hindcast using in situ data, the performance with NWP data such

as GEMLAM24 is less optimal (Figure 6 - Table 9). The validation period covers only one winter, and results indicate that forecasts driven by NWP inputs perform less well than those using ground station data—primarily due to a systematic under-estimation of precipitation by the GEMLAM in the study area (Côté et al., 2017). This limitation may affect the model's ability to anticipate loading events critical for avalanche release. Therefore, while this approach holds operational potential, further multi-year validation and improvement in precipitation amount estimation are necessary to ensure robustness.

Finally, the model's limitations must also be considered. It occasionally failed to predict avalanche occurrences, particularly during warm or rain-on-snow events. These failures highlight the importance of carefully integrating ML outputs into a broader operational workflow that includes human judgment and the promising data-driven assessment (Herla et al., 2025; Pérez-Guillén et al., 2025; Techel et al., 2024).

## 6 Conclusions

This study demonstrates that avalanche occurrence on key routes in the northern Gaspésie region can be effectively predicted by integrating specific meteorological variables with expert-based models, providing valuable insights for avalanche risk management. The inclusion of short-duration, high-intensity precipitation variables, such as snow accumulation over 24-72 hours, aligns with previous findings across various mountainous regions and proves effective for identifying both winter storm-induced avalanches and spring rain-on-snow events. The SnowDriftIndex further enhances predictions for storm slab avalanches, while 450 wind and snow accumulation variables complement each other to account for various storm scenarios.

Our models indicate that while complex algorithms, such as random forests, offer robust predictive power, simpler expert models with a limited set of essential variables (Snow_24h, Snow_72h, Tmean_48h, Rain_24h) remain highly effective and are more accessible for operational management. These models, providing critical thresholds, allow road safety managers to make timely decisions, activate alerts, and deploy resources as needed. The use of machine learning methods like LR and RF 455 models shows that model performance can match or exceed that of current forecasting tools, enhancing hazard anticipation while maintaining a user-friendly framework suitable for real-time application. Our study also demonstrates the potential to use numerical weather prediction to forecast 24 and 48 h in advance avalanche occurrences on the road. However, uncertainty remains in the prediction from the uncertainty in numerical weather prediction.

In light of climate change, which is expected to increase the frequency of wet snow avalanches, integrating temperature 460 and rain variables into predictive models will become increasingly important. Our findings suggest that adapting management protocols in response to evolving weather patterns and snowpack processes will be essential for ensuring the continued safety of road users. Ultimately, the study highlights a comparison for advancing avalanche forecasting by combining statistical models, expert knowledge, and numerical weather prediction. This evaluation has significant potential for broader application in mountainous regions, allowing for data-driven avalanche management that balances model accuracy with practical usability.

*Acknowledgements.* The authors would like to acknowledge Québec's Ministry of Transportation, specifically François Bossé for his supportive work during this project. We would like to thank the members of the LGGRM at UQAR for their support during this long project over the years. We would like to thank Dominic Boucher from Avalanche Québec for providing us with their forecast dataset. Finally, we would like to acknowledge the reviewers Frank Techel, Eric Peitzsch, and Christina Perez for their very helpful and constructive comments that significantly improve our manuscript.

*Disclaimer.* This work was funded mainly by Quebec's Ministry of Transportation (MTMQ) to enhance their risk mitigation program against rock/ice falls and snow avalanches along the roads 132 in northern Gaspésie. Additional funding was provided by the Natural Sciences and Engineering Research Council of Canada (NSERC).

*Author contributions.* FG conceptualized the research project, acquired the funding, assisted JL in the analysis, drafted, and planned the manuscript. JL prepared the dataset, developed the learning workflow and ran the machine learning model. FM assisted JL in the analysis
and wrote the final manuscript.

*Code and data availability.* The dataset of snow avalanche occurrences is the property of Quebec's Ministry of Transportation and could be available upon request for scientific projects. The weather dataset is available on the open-access repository of Environment Canada.

# Appendix A: Feature selection during RFE for model LR and NN

**Table A1.** Cumulative importance (FI) of variables selected by the RFs and count (Count) of variables pre-selected by the RFE and used as input to the LR and NN models for training the models. In bold: variables selected for the expert model.

| Variables | FI | Rank FI | Count | Rank Count | Indice |
|---|---|---|---|---|---|
| Int_snow_12h/24h | 3.15 | 1 | 5 | 8 | 9 |
| SnowDriftIndex_72h | 1.89 | 3 | 7 | 7 | 10 |
| Snow_120h | 1.02 | 9 | 30 | 1 | 10 |
| Int_snow_8h/24h | 1.03 | 8 | 14 | 3 | 11 |
| SnowDriftIndex_24h | 2.72 | 2 | 2 | 11 | 13 |
| **Snow_72h** | **1.35** | **6** | **7** | **7** | **13** |
| Tmax_12h/24h | 1.14 | 7 | 1 | 12 | 9 |
| Int_snow_12h/48h | 0.90 | 12 | 7 | 7 | 19 |
| SnowDriftIndex_96h | 0.79 | 14 | 9 | 5 | 19 |
| WV_Mean_24h | 0.77 | 17 | 16 | 2 | 19 |
| Tmax_4h/48h | 1.01 | 10 | 3 | 10 | 20 |
| Tmean-1d | 0.92 | 11 | 4 | 9 | 20 |
| Int_snow_4h/48h | 0.86 | 13 | 5 | 8 | 21 |
| WV_gust_Max | 0.77 | 15 | 2 | 11 | 26 |
| Int_snow_4h/24h | 0.77 | 16 | 3 | 10 | 26 |
| Snow_48h | 0.68 | 24 | 10 | 4 | 28 |
| Int_snow_1h/24h | 0.74 | 19 | 1 | 12 | 31 |
| Tmax_48h | 0.74 | 20 | 1 | 12 | 32 |
| Tmax_96h | 0.72 | 21 | 2 | 11 | 32 |
| Snow_96h | 0.72 | 22 | 3 | 10 | 32 |
| SnowDriftIndex_120h | 0.71 | 23 | 2 | 11 | 34 |
| **Tmean_48h** | **0.67** | **26** | **3** | **10** | **36** |
| DTR_d_3 | 0.62 | 32 | 9 | 5 | 37 |
| Tmean_72h | 0.63 | 30 | 3 | 10 | 40 |
| Tmax_2d | 0.64 | 29 | 1 | 12 | 41 |
| Tmean_96h | 0.62 | 34 | 4 | 9 | 43 |
| DTR_d_2 | 0.60 | 35 | 4 | 9 | 44 |
| Tmean_2d | 0.53 | 44 | 16 | 2 | 46 |
| Tmin_8h/24h | 0.59 | 36 | 2 | 11 | 47 |
| DTR_d | 0.58 | 39 | 4 | 9 | 48 |
| Tmin_4h/24h | 0.59 | 37 | 1 | 12 | 49 |
| DTR_d_1 | 0.53 | 45 | 8 | 6 | 51 |
| Tmax_1d | 0.57 | 41 | 2 | 11 | 52 |
| Tmax_8h/24h | 0.49 | 47 | 1 | 12 | 59 |
| WD_gust_Max | 0.48 | 48 | 2 | 11 | 59 |
| Tmean_120h | 0.48 | 49 | 1 | 12 | 61 |
| Tmin_24h | 0.42 | 55 | 1 | 12 | 67 |
| Tmean_24h | 0.41 | 57 | 3 | 10 | 67 |
| SnowDriftIndex_48h | 0.40 | 59 | 4 | 9 | 68 |
| Rain_96h | 0.33 | 63 | 2 | 11 | 74 |
| Int_rain_1h/48h | 0.26 | 67 | 1 | 12 | 79 |
| Int_rain_8h/24h | 0.18 | 72 | 1 | 12 | 84 |
| Int_snow_1h/48h | 1.50 | 4 | #N/A | #N/A | #N/A |
| **Snow_24h** | **1.38** | **5** | #N/A | #N/A | #N/A |

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
