# Peer review of "Assessing the predictive capability of several machine learning algorithms to forecast snow avalanches using numerical weather prediction model in eastern Canada."

_EGUsphere, 2025_

## Referee Comment (RC1)

**Review Gauthier et al. (2025): Assessing the predictive capability of several machine learning algorithms to forecast snow avalanches using numerical weather prediction model in eastern Canada**

Frank Techel

WSL Institute for Snow and Avalanche Research SLF, Davos, Switzerland

**1   Summary**

The study evaluates and compares the predictive performance of four machine learning algorithms for snow avalanche forecasting along a highway corridor in eastern Canada. Models are trained on meteorological data from a local weather station and tested both retrospectively (hindcast) and using 24–48 h numerical weather prediction (NWP) data. The authors also develop simplified "expert models" based on a small set of selected variables and show that these models can match the performance of more complex approaches. They conclude that such models are suitable for operational forecasting and hazard management using NWP data.

   While the manuscript is generally well structured and supported by mostly clear figures and tables, several methodological choices reduce the transparency and interpretability of the results. Key concerns include the definition and weighting of the event variable, inconsistent reporting of dataset size and time coverage, the rationale for mapping model probabilities to danger levels, and limited validation based on a single winter season. Additionally, the forecasting goal and performance metrics are not clearly aligned with the operational context. These issues are outlined in detail below, with the intention of strengthening the manuscript's clarity, methodological robustness, and applied relevance.

**2   Major concerns**

**2.1   Size and composition of event data set**

The manuscript does not report the total number of days included in the dataset used for model development (though the cumulative number of events since 1987 is mentioned), nor the number of event days used for training and testing, or how many days were covered by avalanche forecasts. This lack of detail makes it difficult to evaluate the size, coverage, and representativeness of the training and validation datasets. Please provide these numbers. Additionally, I recommend including a histogram or table showing the distribution of intervention counts per day. If available, it would be helpful to show, in the same figure, the total number of avalanches per day (including those not classified as events), to help readers understand how

often avalanche conditions were favorable for avalanche release and, among those, how often avalanches reached the road. This would also provide the link to Figure 4, where both interventions and avalanches, which didn't reach the road, are shown.

**2.2 Event weighting**

The current approach assigns weights to event days based on the number of interventions per day, duplicating rows with weight factors from 1 to 4. While this may aim to emphasize high-impact days in the dataset, the scaling appears arbitrary. The manuscript would benefit from a clear justification for this specific weighting scheme, including a discussion of whether it more reliably captures avalanche activity than alternative approaches — such as duplicating entries based on the actual number of interventions, using no weighting at all, or treating severity as a separate variable. Additionally, the potential for overfitting introduced by this duplication strategy should be addressed.

Potentially, the following comment concerns this weighting scheme: in Tables 6 and 7, the term "train non double" appears as a label for model results, but it is never defined in the text. Presumably, it refers to training on the non-duplicated (unweighted) dataset, but this should be clearly stated. Please define this term explicitly in the methods section and explain how the results under this label differ in training setup from the standard "train" case with duplicated rows.

**2.2.1 Mapping forecast probabilities to danger levels**

In Section 3.2 and Table 2, the authors introduce a mapping of model-predicted event probabilities to avalanche danger levels. However, the rationale for this mapping remains unclear. It appears that fixed probability ranges (e.g., 50–75% for "Considerable") are assigned to match categorical danger levels, yet it is not explained whether this mapping was derived from empirical data, optimized using forecast performance, or adopted from an existing standard. Shouldn't the probability ranges be obtained following analysis to be in line with the data?

Moreover, it is unclear whether the proposed scale is consistent with the North American Avalanche Danger Scale (Statham et al., 2010), or whether the four-level version used here, and their descriptions which clearly relate to road level avalanche conditions, follow an accepted operational standard, whether this is the definition used by road authorities, or whether it is a custom adaptation. Please clearly state so, justify the choice, and reasoning for this descriptive definition.

Please also provide the respective event and non-event data with the forecast danger levels, i.e., the number of avalanche days to all days for each level, and the average number of events per intervention day.

**2.2.2 Forecasting goal, performance metrics, and contextualisation**

The manuscript lacks a clearly stated forecasting goal — for example, whether the priority is minimizing false negatives (missed events) or false positives (false alarms) or whether a balanced performance is of interest (Sect. 3.4.6). While F1 score and AUC are widely used, the authors do not critically discuss whether these metrics allow to reflect meaningful operational gains over current forecasting practice. In particular, the F1 score assumes equal cost for false positives and false negatives, which may not be appropriate in avalanche forecasting where missing a high-impact event can have serious consequences.

The claim (L359) that the models are efficient and suitable for supporting road safety management would be stronger if it were grounded in such operational considerations. Ebert and Milne (2022) provides an in-depth discussion of these metrics for evaluating rare-and-severe event forecasts may be useful.

**2.2.3 Inconsistent and limited model validation across time periods**

It is somewhat confusing — or perhaps just not clearly explained — that results from datasets with differing temporal coverage are directly compared (L261–271): the hindcast includes only a single season (2019), the Test dataset spans five seasons (L156), and the Avalanche Québec forecasts are evaluated over either five or seven seasons (Table 8 refers to 2013–2020, while L156 defines five seasons). Please clarify which time periods are used for each comparison, and ensure consistent reporting of performance metrics across these datasets.

Moreover, while presenting a single season (2019 in Figure 4) as an illustrative example is appropriate, using it as the only hindcast validation case is insufficient to evaluate model robustness across varying snow and weather conditions. As shown in Table 7, only nine event days occurred during this season, which limits the strength of conclusions about model performance (Section 5.2). Please justify the selection of this specific year for hindcast validation and discuss the implications this has for the operational relevance and generalizability of the findings.

**3 Minor concerns**

– Study area: Please provide the elevation of the Cap-Madeleine weather station and the elevation range of the avalanche release areas. What is the elevation of the GEMLAM grid cell used in the analysis (L186–187)? - This information is essential to assess the representativeness of the input data and the applicability of the forecasts to release zone conditions. In case elevation differs, take this up in the Discussion. Also, in the discussion, address that some road sections lie nearly 100 km from the weather station, which may limit forecast reliability in parts of the forecast area.

– **Section 3.2:** The manuscript uses regional avalanche forecasts issued by Avalanche Québec as a reference, but it is not clear why these were chosen over road-specific hazard assessments. Please clarify their relevance in the study context. In this regard, it may be helpful to include more detail on the operational hazard assessments carried out by road authorities — possibly in the Study Area section. For instance: Are roads preventively closed? Are Avalanche Québec's forecasts intended for, and actively used by, road authorities in their decision-making? When interpreting the results, consider discussing how the avalanche forecasts relate to actual road closures or missed closures (e.g., in terms of precision, false alarm rate, or true positive rate), as this would provide a clearer link to the intended operational application and highlight the challenges in forecasting avalanches that reach the road.

– Figure 1: Please make the location of the Cap-Madeleine weather station more prominent - e.g., using a larger dot or different color.

– **L129:** The notation "1h/24, 4h/24, 8h/24, 12h/24, 24h/24, 1h/48, 4h/48, 8h/48, 12h/48, 24h/48, 48h/48, 72h/72, 96h/96, 120h/120" is not intuitively clear. Additionally, Table 3 lists different values for $int_{rain}$ and $N_{\mathrm{hr,snow}}$. Please add explanatory detail, perhaps moving this notation to Table 3 where similar forms are used for temperature variables.

– Table 3: There are several inconsistencies and typos, such as the use of French ("et") instead of English, and mismatched or unclear time intervals. For example, $N_{\mathrm{snow}}$ is described as 24 h on the left but listed as 24 and 48 h on the right. Why are $N_{\mathrm{snow}}(24h)$ and $N_{\mathrm{snow}}(48h)$ grouped on one line, while longer aggregations appear separately? Please revise for consistency and clarity.

– Table 3 and Results section: Is $N_{\mathrm{hr,snow}}$ (Table 3) the same as $Snow_24h$ and $Snow_72h$ (L304). - Please ascertain that all abbreviations are used consistently throughout.

– Section 3.4.1: This section appears to mix the introduction of NWP data with the process of selecting expert model variables. Please introduce the NWP data in a separate paragraph—either as a dedicated subsection (e.g., Section 3.4) or by restructuring Section 3.3 into subsections for weather station data, NWP data, and derived variables. Additionally, clarify how expert judgment guided variable selection beyond redundancy (step 1), especially since the variables in Table 6 differ significantly from those ultimately selected (L251). Consider dedicating a specific section to this.

– Figure 4: A scale or reference for avalanche activity is missing. It would be helpful to plot the number of interventions and non-intervention avalanche observations above the activity bars. Also, please clarify whether the activity bars represent a histogram or another form of count aggregation.

– Section 4.3: please state whether the "expert models" are analyzed

– **L299–305:** The SnowDriftIndex is discussed as an additional variable of interest, but no results are shown regarding its impact or inclusion in model evaluations. Please report its contribution to performance to ground this discussion in empirical evidence.

**References**

Ebert, P. A. and Milne, P.: Methodological and conceptual challenges in rare and severe event forecast-verification, Nat. Hazards Earth Syst. Sci., 22, 539–557, https://doi.org/10.5194/nhess-22-539-2022, 2022.

Statham, G., Haegeli, P., Birkeland, K., Greene, E., Israelson, C., Tremper, B., Stethem, C., McMahon, B., White, B., and Kelly, J.: The North American public avalanche danger scale, in: Proceedings ISSW 2010. International Snow Science Workshop, 17 - 22 Oct, Lake Tahoe, Ca., pp. 117–123, 2010.

---

## Referee Comment (RC2)

Review of *Assessing the predictive capability of several machine learning algorithms to forecast snow avalanches using numerical weather prediction model in eastern Canada.* Gauthier et al. (egusphere-2025-1572)

**General Comments**

In this study, the authors use four different machine learning (ML) techniques for predicting avalanche events impacting a road along the north shore of the Gaspé Peninsula, Quebec, in eastern Canada. They trained the models using 70% of the data from 2003/04 to 2012/13 and tested the data on five winters 2015/16 to 2019/20. In addition to using a suite of 80+ variables, the authors also used a limited number of variables in an 'expert' model. They also assessed the 24 hour predictive capability of these ML techniques using numerical weather products (NWP). The authors detail the accuracy of each specific ML technique using all variables, expert model, and NWP forced regimes. They found the 'expert' models were highly effective and potentially more accessible for forecasting operations.

Overall, the manuscript is generally well written and organized. The dataset appears suitable for this type of analysis, and the methods are appropriate and reasonable. However, there are several major topics the authors should consider. Here, I provide a few general comments and then specific comments below.

1.  The first two major issues pertain to data applicability and sample size reporting. The authors use data from ten winter seasons to train the data and then from five winters to test the data. Given non-stationarity of meteorological and climate variables as well as interannual variability, how well do the test data represent the training dataset and vice-versa. In other words, are you testing the model on winters similar enough to the winters on which they were trained? How might using time series data with the potential of non-stationarity influence your results? Can you provide some measure of variability and spread that shows these two winter datasets are reasonable for training and testing? Additionally, when assessing the 2019 season, was the season similar to the test data seasons? Given the decent scores, it appears so, However, when making recommendations for an operation, it is important to note that different variables or interactions of variables different than the training dataset could still result in avalanches resulting in poor model performance (Peitzsch et al., 2012).

2.  You state the number of avalanches (n=861) over 153 event days from 1987 to 2020, but not for the period of record you studied. From the confusion matrix it looks like n=209 for the training dataset but n=489 for LR and NN, but n=479 for CT and RF for the test data (Table 6). 1) Why are there over 2x as many event days in the test data compared to the training data when there are 2x as many years in the test data? This seems peculiar. If this is indeed the case, please provide background in the main text. 2) Why are there two different values (n=479 and n=489) for test data in Table 6? It would be helpful to state the sample size overall value of avalanches and event days for the study period, the sample size for $E_{AV A}$ during this period, and the sample size for the training and test datasets in the main text.

    Additionally, the weighting for different number of interventions per day seems rather arbitrary. Can you provide more context for using this metric as opposed to using perhaps a modification of the Avalanche Activity Index (Schweizer et al., 2003) that accounts for size. Accounting for size might also allow you to include avalanches impacting the road as well as those that don't.

Forecasters are certainly considering avalanche activity that doesn't reach the object at risk and this might provide operations a continuum of results on which to base their decision.

3. The comparison of ML output probability with Avalanche Québec's danger levels is a bit of a mismatch of scale. As you state, avalanche operations are incorporating more statistical and physically based snowpack models into their workflow. However, they are likely (hopefully) using scale appropriate simulations or datasets. In your study, the models are trained to predict avalanches that reach Road 132, a very specific local area. The danger ratings are a regional hazard forecast, not necessarily a true reflection of avalanche occurrence for that given day. It would be more useful to compare ML predictive capacity to avalanche occurrence (of similar size) throughout the region rather than danger ratings themselves, since the forecasted danger level also contains an unknown amount of uncertainty. Please discuss the rationale for choosing danger ratings and limitation therein.

4. There is currently very little to no mention of limitations and potential sources of uncertainty or error in the manuscript. Given your suggestion to use ML for operational forecasting workflows, it seems necessary to include a section that details the limitations in this study. It would also be useful to include potential considerations or statements of caution if avalanche forecasting operations are contemplating using ML model output for decision-making.

5. As the authors point out, ML techniques have been used to study avalanche occurrence and improve avalanche forecasting. The use of these techniques is not novel. The addition of a comparison with a more parsimonious 'expert' model, the use of a NWP, and the integration of public avalanche forecasts for validation make this paper a worthy contribution to the existing body of literature. However, some of the Discussion reads like a consultant report for a specific client rather than a manuscript for the broader scientific community. For example, there are more suggestions for this specific forecasting operation than broad comparisons with other studies that might mike this manuscript more applicable to other operations. Please expand upon this in the Discussion. Additionally, there is a lot of content in the Discussion devoted to using the model in operational contexts. It would be helpful to mention how incorporating these results/ML output differs from what the operation currently uses. In other words, how would things have improved (or not) if they used this model versus what they currently do? You allude to this in lines 373-375, but providing further evidence of how previous assessments would have been improved could be helpful here. It would also be useful to describe the days the models failed to predict events since the consequences would be more severe.

**Specific and Technical Comments**
Abstract: please write/define all abbreviations (e.g. MTMQ, LR, and RF)

Line 24: spreading zone? Do you mean runout or deposition zone?

Line 29: 'windy' remove 'y'

Line 84/Figure 3: This is a bit confusing. The 'Data from NWP model' arrow makes it look like this was used to create the expert model variables. I thought the expert model variables were manually selected. Also, where is the expert model in this flow chart?

Line 85/Sec. 3.1: See comment above re: sample size values. It is necessary to know the sample size for training and test datasets in the text for proper evaluation.

Line 91: I don't understand how $E_{AV\,A}$ is a binary variable. It seems there are 4 possible values (1-4) used to weight if the binary value of avalanche hitting the road (1) or not (0) is 1. The variables simply is a weighting factor. Please clarify.

Line 109-110: The current wording is confusing because the probability of an avalanche reaching the road, as determined by the models, can also be less than 50%. Do you mean 'All probabilities above 50% were classified as event days.'? Also, change 'Considerate' to 'Considerable'.

Line 140: BD should be DB, I assume, for database. Either define DB first or use full word.

Line 171: Seems like you need a citation that demonstrates that model output can be daunting.

Line 193: Change TC to CT.

Line 197: CA should be CT.

Line 212: 'data is processed' to 'data are processed'

Line 226: 'unbalanced datasets'. Since you balanced the training datasets (Sec. 3.1), I assume you mean 'unbalanced test datasets' as per the next sentence. Consider clarifying this here.

Table 6,7,8: Please define acronyms for the reader in the caption. Also define Matrice (i.e. confusion matrix).

Line 261: See general comment above re: representativeness of assessing model with just one season.

Line 314-319: It would be helpful to know the proportion of events classified as wet (wet loose, wet slab, or glide) in both the training and test datasets. You are simply including these variables because wet avalanches are likely to become more frequent, but the model is trained on historical data.

Line 321: Use caution when suggesting practitioners can use specific threshold values for forecasting purposes. See general comment above regarding limitations. There is some uncertainty in the models and there should be some confidence intervals surrounding these values if recommending their use. Using specific model-derived values as a threshold for operational decision making can be problematic, particularly in the context of non-stationarity.

References
Peitzsch, E. H., Hendrikx, J., and Fagre, D. B.: Timing of wet snow avalanche activity: an analysis from Glacier National Park, Montana, USA, Proceedings of the 2012 Interational Snow Science Workshop, Anchorage, Alaska, USA, 884-891, 2012.

Schweizer, J., Kronholm, K., and Wiesinger, T.: Verification of regional snowpack stability and avalanche danger, Cold Regions Science and Technology, 37, 277-288, 10.1016/s0165-232x(03)00070-3, 2003.

---

## Referee Comment (RC3)

**Assessing the predictive capability of several machine learning algorithms to forecast snow avalanches using numerical weather prediction model in eastern Canada (egusphere-2025-1572)**

This paper presents several machine learning models for predicting avalanche events in eastern Canada. The study addresses an emergent topic in avalanche forecasting, with increasing demand for developing data-driven models that can be applied in practice. The authors used typical machine learning models, such as Logistic Regression (LC), Tree classification (TC), Random Forests (RF) model, and an Artificial Neural Network (NN), and compared the variation of the performance using variables extracted from meteorological data, local avalanche observations, and avalanche forecasts of Québec region. While the content, structure, and results of the paper are good and relevant to the topic, improvements are needed in the data used to develop the models, the definition of the target variable, and the detailed explanation of the methods and strategies employed. I recommend publishing this paper after the authors have addressed the following comments and suggestions. Please see my detailed feedback below.

**General comments:**

1. It would be very useful to add more detailed information on the types of avalanches typically released in the area, their locations (including typical paths in the map of Figure 1), and how these data and observations are collected. Are avalanches observed daily? Are the observations human-based? How accurately is the avalanche release time known? What happens during snowstorms? Is the road closed during periods of high avalanche danger? How far is the meteorological station from the avalanche release areas? Specifically, I recommend adding much more information about the type, size, and release time of the avalanche events in the dataset described in Section 3.1. Additionally, making this dataset open source would be beneficial for other studies.

2. I suggest defining the target variable more clearly in Table 1 and the rest of the manuscript. For example, Eava = 'avalanche day' / 'no avalanche day'. This definition should be used consistently in all the confusion matrices of the tables. Also, to qualify as an avalanche day, what are the requirements? For example: at least one avalanche of size X (what is the minimum size or path length to reach the road?) Additionally, the selection of the weights in Table 1 is not justified or supported by previous studies.

3. Given that the meteorological drivers for dry and wet snow avalanches differ, are the models developed to predict dry avalanches, wet avalanches, or both? This is only mentioned in the Discussion section and could be earlier in the manuscript.

4. The explanation of how the data is merged and the target variables used to test the hindcast and forecast models can be explained in more detail. What is the time window of the avalanche forecasts from Avalanche Québec? How is this merged with the variables extracted from the time series of meteorological data and avalanche activity (point data)? The release time is very important when correlating with the meteorological variables used to develop the models. For instance, if an avalanche is released at 01:00, daily meteorological variables computed from 00:00 to 00:00 of the following day would not accurately represent that event.

5. Could you please provide more information about the avalanche forecast issued in Québec? When is this forecast released, and what is the valid time window of the bulletin?

6. Why were these probability thresholds assigned to each danger level? It would be helpful to show the distribution of all the avalanche and non-avalanche events in relation to the danger level forecasts in Québec. This type of analysis could be used to justify or derive the chosen probability thresholds. Additionally, according to the European Avalanche Danger Scale, danger levels 1 and 2 do not imply the absence of avalanche activity and are dependent on avalanche size (1; 2). For instance, the time series results in Figure 4 show that 4 out of a total of 9 avalanches that reached the road occurred when the forecast danger level was 1 or 2.

7. Feature selection: It would be interesting to provide more information about the feature selection approach. For example, including a correlation matrix in the Appendix. Since Recursive Feature Elimination requires as input the number of input variables, it would be good to show how varying this number affects the performance of the LR and NN models. Does the final selection correspond to the number of features that yields the maximum F1-score?

8. Data splitting and balancing: You mention that splitting the data into training and test sets with a 70/30 ratio is optimal, and that the avalanche and no-avalanche events were balanced to train the models. It would be helpful to include, for example, a bar plot showing the total distribution of events for each class and their distribution in the training and test sets. The confusion matrices in Tables 6 and 7 do not reflect a balanced dataset. Please check my specific comments on these tables below.

9. I suggest moving the description of the machine learning models and the definition of the evaluation metrics to the Appendix. Instead, the main text could focus more on the final model selected, including details such as the architecture of the neural network and the hyperparameters used for the other models. Additionally, providing the code and data used to develop the models would improve the reproducibility of the study and be valuable for future research.

10. For evaluating the performance of the hindcast and forecast models (GEMLAM 24h and 48h) shown in Table 7, what "ground truth" is used to evaluate the performance? Are the danger level forecasts merged by combining levels 1 and 2 as "non-avalanche" events and levels 3 and 4 as "avalanche" events? This should be clearly stated in the caption of the tables and the main text. I am not convinced that merging danger levels 1 and 2 as non-avalanche and levels 3 and 4 as avalanche events is appropriate for defining the ground truth, especially since the actual ground truth in this case consists of observed avalanche events. Instead, presenting a correlation analysis between the model outputs and the avalanche forecasts may be more suitable.

**Specific comments:**

- Figure 1: It is difficult to see the location of the Cap-Madeleine weather station (the legend should be also translated to English). It would be very helpful to show the outlines or locations of the typical avalanche paths used in this study on the map.

- Lines 61–62: The references to Figure 1 are incorrect; they should refer to Figure 2.

**Tables 6 and 7:**

- The captions of these tables could be improved by including more detailed information.

- The differences between the results shown in the two tables are not clearly explained. Table 6 presents the performance of the models trained with a different set of variables after the feature selection process, right? Table 7 shows the performance of each model using only four variables. Why does the CT model use only three variables?

- The terms "train" and "train non double" are not defined in the main text or in the captions of the tables. Although it is mentioned that the models have been balanced (Line 158), the "train" dataset is not balanced, as shown in the confusion matrices of both Tables. Additionally, both "train" and "train non double" appear to contain significantly fewer samples than the test set. It is unclear whether the results for the test set were obtained using the "train" or "train double" datasets.

- The reason for showing the performance on these two training sets is also unclear, as their results are not discussed in the text. It is assumed that the results are based on 50 repetitions of the models (Line 160). If that is the case, are the reported scores the averages over those 50 repetitions? If so, a different confusion matrix should be obtained for each repetition.

- There is an inconsistency in the count of avalanche events reported in Table 6: the test set contains 42 events for the LR and NN models, while only 32 events for the other models. Also, the event counts in Table 7 for the NN model differ from the rest.

**Table 8:**

- The captions of these tables could be improved by including more detailed information.

- Which model is used here?

**References**

1. EAWS. European Avalanche Danger Scale. https://www.avalanches.org/standards/avalanche-danger-scale/ (2021). [Online; last access 22-May-2025].

2. Schweizer, J., Mitterer, C., Techel, F., Stoffel, A. & Reuter, B. On the relation between avalanche occurrence and avalanche danger level. *The Cryosphere* **14**, 737–750, DOI: 10.5194/tc-14-737-2020 (2020).

---

## Author Comment (AC1)

**Response to reviewer's comments**

Manuscript EGUSPHERE-2025-1572

Assessing the predictive capability of several machine learning algorithms to forecast snow avalanches using numerical weather prediction model in eastern Canada

by Gauthier, Laliberté and Meloche

In the following, we provide (in blue) detailed point-by-point answers to the comments raised by the reviewers (in black, *italic*). In addition, modifications made to the manuscript are highlighted in a separate file with tracked-changes.

**Response to Referee #1– Dr. Frank Techel**

**1  Summary**

*The study evaluates and compares the predictive performance of four machine learning algorithms for snow avalanche forecasting along a highway corridor in eastern Canada. Models are trained on meteorological data from a local weather station and tested both retrospectively (hindcast) and using 24–48 h numerical weather prediction (NWP) data. The authors also develop simplified "expert models" based on a small set of selected variables and show that these models can match the performance of more complex approaches. They conclude that such models are suitable for operational forecasting and hazard management using NWP data. While the manuscript is generally well structured and supported by mostly clear figures and tables, several methodological choices reduce the transparency and interpretability of the results. Key concerns include the definition and weighting of the event variable, inconsistent reporting of dataset size and time coverage, the rationale for mapping model probabilities to danger levels, and limited validation based on a single winter season. Additionally, the forecasting goal and performance metrics are not clearly aligned with the operational context. These issues are outlined in detail below, with the intention of strengthening the manuscript's clarity, methodological robustness, and applied relevance.*

**2  Major concerns**

**2.1  Size and composition of event data set**

*The manuscript does not report the total number of days included in the dataset used for model development (though the cumulative number of events since 1987 is mentioned), nor the number of event days used for training and testing, or how many days were covered by avalanche forecasts. This lack of detail makes it difficult to evaluate the size, coverage, and representativeness of the training and validation datasets. Please provide these numbers. Additionally, I recommend including a histogram or table showing the distribution of intervention counts per day. If available, it would be helpful to show, in the same figure, the total number of avalanches per day (including those not classified as events), to help readers understand how often avalanche conditions were favorable for avalanche release and, among those, how often avalanches reached the road. This would also provide the link to Figure 4, where both interventions and avalanches, which didn't reach the road, are shown.*
We thank Referee #1 for their positive comments on our manuscript. To address this comment, we have added a figure with two subplots as suggested. The first subplot describes the number of event days and interventions for each year throughout our dataset (both training and test datasets). The second subplot shows the frequency of the number of avalanches per day, for both training and test datasets. Unfortunately, we cannot show the avalanches that did not reach the road, as the observations were only taken after 2015.

**2.2  Event weighting**

*The current approach assigns weights to event days based on the number of interventions per day, duplicating rows with weight factors from 1 to 4. While this may aim to emphasize high-impact days in the dataset, the scaling appears arbitrary. The manuscript would benefit from a clear justification for this specific weighting*

*scheme, including a discussion of whether it more reliably captures avalanche activity than alternative approaches — such as duplicating entries based on the actual number of interventions, using no weighting at all, or treating severity as a separate variable.* In order to demonstrate the selection of the scaling of the weights, we added a figure (Figure 4.b) showing the distribution of the number of avalanches (interventions) per day, with a histogram with the associated bins used for the weights.

*Additionally, the potential for overfitting introduced by this duplication strategy should be addressed. Potentially, the following comment concerns this weighting scheme: in Tables 6 and 7, the term "train non double" appears as a label for model results, but it is never defined in the text. Presumably, it refers to training on the non-duplicated (unweighted) dataset, but this should be clearly stated. Please define this term explicitly in the methods section and explain how the results under this label differ in training setup from the standard "train" case with duplicated rows.* The results from the non-duplicated dataset (non-double) were removed to improve clarity, presenting only the train dataset results. The potential overfitting issue from the duplication strategy is addressed in a new discussion section about bias and uncertainty (also suggested by reviewer 2).

**2.2.1 Mapping forecast probabilities to danger levels**

*In Section 3.2 and Table 2, the authors introduce a mapping of model-predicted event probabilities to avalanche danger levels. However, the rationale for this mapping remains unclear. It appears that fixed probability ranges (e.g., 50–75% for "Considerable") are assigned to match categorical danger levels, yet it is not explained whether this mapping was derived from empirical data, optimized using forecast performance, or adopted from an existing standard. Shouldn't the probability ranges be obtained following analysis to be in line with the data?* The essential goal was to compare the forecasted danger level (traditional forecasting) to the model probabilities. However, as pointed out by all the reviewers, we realized that this association is rather arbitrary. It is difficult to derive thresholds of probabilities because the model predicts the probability of a single event happening compared to the danger level "High" where multiple avalanches are expected. We will remove this section from the methods and only visually compare the model predictions with the danger level forecasts by AvQc. We will only assess the true prediction of the avalanche forecast by AvQc at Considerable level (or High) where an avalanche is expected on the road and compute the performance metrics. This will allow us to compare traditional forecasting with model performance.

*Moreover, it is unclear whether the proposed scale is consistent with the North American Avalanche Danger Scale (Statham et al., 2010), or whether the four-level version used here, and their descriptions which clearly relate to road level avalanche conditions, follow an accepted operational standard, whether this is the definition used by road authorities, or whether it is a custom adaptation. Please clearly state so, justify the choice, and reasoning for this descriptive definition. Please also provide the respective event and non-event data with the forecast danger levels, i.e., the number of avalanche days to all days for each level, and the average number of events per intervention day.* We clarify that the decision to merge danger levels 4 and 5 was made by Avalanche Québec forecaster's team and the local direction of Québec's Ministry of Transport.

**2.2.2 Forecasting goal, performance metrics, and contextualisation**

*The manuscript lacks a clearly stated forecasting goal — for example, whether the priority is minimizing false negatives (missed events) or false positives (false alarms) or whether a balanced performance is of interest (Sect. 3.4.6). While F1 score and AUC are widely used, the authors do not critically discuss whether these metrics allow to reflect meaningful operational gains over current forecasting practice. In particular, the F1 score assumes equal cost for false positives and false negatives, which may not be appropriate in avalanche forecasting where missing a high-impact event can have serious consequences. The claim (L359) that the models are efficient and suitable for supporting road safety management would be stronger if it were grounded in such operational considerations. Ebert and Milne (2022) provide an in-depth discussion of these metrics for evaluating rare-and-severe event forecasts that may be useful.* We changed the sentences in the subsection 3.4.6 to state our forecasting objective: "However, the forecasting objective is to find the best compromise between the number of detections (motorist safety) and false alarms (additional operating costs). Thus, maximizing the number of events predicted (Precision or True Positive Rate TPR) while minimizing false alarms (False

Alarm Rate FAR or False Positive Rate FPR). Improvement in one often leads to deterioration in the other. The Receiver Operating Characteristic (ROC) and $F_1$ make it possible to establish a relationship between predicted events (Prec or TPR) and false alarms (FPR or FAR)."

**2.2.3 Inconsistent and limited model validation across time periods**

*It is somewhat confusing — or perhaps just not clearly explained — that results from datasets with differing temporal coverage are directly compared (L261–271): the hindcast includes only a single season (2019), the Test dataset spans five seasons (L156), and the Avalanche Québec forecasts are evaluated over either five or seven seasons (Table 8 refers to 2013–2020, while L156 defines five seasons). Please clarify which time periods are used for each comparison, and ensure consistent reporting of performance metrics across these datasets. Moreover, while presenting a single season (2019 in Figure 4) as an illustrative example is appropriate, using it as the only hindcast validation case is insufficient to evaluate model robustness across varying snow and weather conditions. As shown in Table 7, only nine event days occurred during this season, which limits the strength of conclusions about model performance (Section 5.2). Please justify the selection of this specific year for hindcast validation and discuss the implications this has for the operational relevance and generalizability of the findings.* We have included details on the validation across different time periods in the revised manuscript. The time frames of the *Train* and *Test* datasets are now clearly depicted in Figure 4a, as well as in Tables 6 and 7. The *Test* dataset functions as a hindcast validation including all winter days, covering five winters. Revisions have been made to Tables 6 and 7 for improved clarity, along with updates to the text in Section 3.4.6 (Performance Indicators). Additionally, Table 7 has been divided into two separate tables: the first table compares only the expert models' results for the train and test datasets (hindcast spanning 5 years), and the second table (now Table 9) contrasts the specific hindcast of winter 2019 with the GEMLAM24-48h for that particular winter.

**3 Minor concerns (Reviewer 1)**

- *Study area: Please provide the elevation of the Cap-Madeleine weather station and the elevation range of the avalanche release areas. What is the elevation of the GEMLAM grid cell used in the analysis (L186–187)? - This information is essential to assess the representativeness of the input data and the applicability of the forecasts to release zone conditions.* The altitude of the weather station (20 m) and the GEMLAM grid altitude (11 m) has been added to the text.

- *In case elevation differs, take this up in the Discussion. Also, in the discussion, address that some road sections lie nearly 100 km from the weather station, which may limit forecast reliability in parts of the forecast area.* We added a few sentences in a new discussion section (5.3) about bias and uncertainty. We addressed in this section the representativity of the weather station, especially the west sector which is nearly 60 km away.

- *Section 3.2: The manuscript uses regional avalanche forecasts issued by Avalanche Québec as a reference, but it is not clear why these were chosen over road-specific hazard assessments. Please clarify their relevance in the study context. In this regard, it may be helpful to include more detail on the operational hazard assessments carried out by road authorities–possibly in the Study Area section. For instance: Are roads preventively closed? Are Avalanche Québec's forecasts intended for, and actively used by, road authorities in their decision-making? When interpreting the results, consider discussing how the avalanche forecasts relate to actual road closures or missed closures (e.g., in terms of precision, false alarm rate, or true positive rate), as this would provide a clearer link to the intended operational application and highlight the challenges in forecasting avalanches that reach the road.* More details were added in section 3.2 about the specific forecast and road closure, as this was also suggest by reviewer 3.

- *Figure 1: Please make the location of the Cap-Madeleine weather station more prominent - e.g., using a larger dot or different color.* Done

- *L129: The notation "1h/24, 4h/24, 8h/24, 12h/24, 24h/24, 1h/48, 4h/48, 8h/48, 12h/48, 24h/48, 48h/48, 72h/72, 96h/96, 120h/120" is not intuitively clear. Additionally, Table 3 lists different values for intrain*

and Nhr,snow. Please add explanatory detail, perhaps moving this notation to Table 3 where similar forms are used for temperature variables. *The notations and captions were modified and improved for more clarity in Table 3.*

- *Table 3: There are several inconsistencies and typos, such as the use of French ("et") instead of English, and mismatched or unclear time intervals. For example, Nsnow is described as 24 h on the left but listed as 24 and 48 h on the right. Why are Nsnow(24h) and Nsnow(48h) grouped on one line, while longer aggregations appear separately? Please revise for consistency and clarity.* *The inconsistencies in typos were removed and, we used Snow now throughout the manuscript. We added in Table 3, a line stating that Snow24-48 are not in the same line because they are not in the same correlation group for RFE.*

- *Table 3 and Results section: Is Nhr,snow (Table 3) the same as Snow24h and Snow72h (L304). - Please ascertain that all abbreviations are used consistently throughout.* *Corrected*

- *Section 3.4.1: This section appears to mix the introduction of NWP data with the process of selecting expert model variables. Please introduce the NWP data in a separate paragraph—either as a dedicated subsection (e.g., Section 3.4) or by restructuring Section 3.3 into subsections for weather station data, NWP data, and derived variables.* *The paragraph about NWP was moved to the meteorological section above.* *Additionally, clarify how expert judgment guided variable selection beyond redundancy (step 1), especially since the variables in Table 6 differ significantly from those ultimately selected (L251). Consider dedicating a specific section to this.* *The sentence was modified to incorporate our reflections essentially for both avalanche type (dry and wet) with 1) loading component by rain or snow, and 2) temperature for melting: "our understanding from previous work on the development of instabilities in the snowpack for both dry and wet avalanches in the study area (Gauthier et al., 2017), ultimately leading us to snow/rain accumulation (loading), and temperature (melting) (Ancey, 2006; McClung and Schaerer, 2006)."*

- *Figure 4: A scale or reference for avalanche activity is missing. It would be helpful to plot the number of interventions and non-intervention avalanche observations above the activity bars. Also, please clarify whether the activity bars represent a histogram or another form of count aggregation.* *The number of avalanches was already in the figure but not very visible, we increase the size of the bar plot (avalanche count), and decrease others elements, while also adding a dedicated secondary y axis for the number of avalanches on and off the road.*

- *Section 4.3: please state whether the "expert models" are analyzed* *Done*

- *L299–305: The SnowDriftIndex is discussed as an additional variable of interest, but no results are shown regarding its impact or inclusion in model evaluations. Please report its contribution to performance to ground this discussion in empirical evidence.* *The results of the Snowdrift index are presented in Table 6 and at line 253-255. The variable Snowdrift_24h was only selected by two out of the four ML models (LR and NN) as a significant predictor of the event variable.*

**References**

Ancey, C.: Dynamique des avalanches, PPUR presses polytechniques, 2006.

Ebert, P. A. and Milne, P.: Methodological and conceptual challenges in rare and severe event forecast verification, Natural Hazards and Earth System Sciences, 22, 539–557, https://doi.org/10.5194/nhess-22-539-2022, 2022.

Gauthier, F., Germain, D., and Hétu, B.: Logistic models as a forecasting tool for snow avalanches in a cold maritime climate: northern Gaspésie, Québec, Canada, Natural Hazards, 89, 201–232, https://doi.org/10.1007/s11069-017-2959-3, 2017.

McClung, D. and Schaerer, P.: The avalanche Hanbook, The Mountaineers Books, 2006.

Statham, G., Haegeli, P., Birkeland, B., Greene, E., Israelson, C., Tremper, B., Stethem, C., McMahon, B., White, B., and Kelly, J.: The North American public avalanche danger scale., in: International Snow Science Workshop, Squaw Valley, pp. 117–123, Squaw Valley, 2010.

---

## Author Comment (AC2)

**Response to reviewer's comments**

Manuscript EGUSPHERE-2025-1572

Assessing the predictive capability of several machine learning algorithms to forecast snow avalanches using numerical weather prediction model in eastern Canada

by Gauthier, Laliberté and Meloche

In the following, we provide (in blue) detailed point-by-point answers to the comments raised by the reviewers (in black, *italic*). In addition, modifications made to the manuscript are highlighted in a separate file with tracked-changes.
* * *
**Response to Referee #2– Dr. Erich Peitzsch**

**1    General Comments**

*In this study, the authors use four different machine learning (ML) techniques for predicting avalanche events impacting a road along the north shore of the Gaspé Peninsula, Quebec, in eastern Canada. They trained the models using 70% of the data from 2003/04 to 2012/13 and tested the data on five winters 2015/16 to 2019/20. In addition to using a suite of 80+ variables, the authors also used a limited number of variables in an 'expert' model. They also assessed the 24 hour predictive capability of these ML techniques using numerical weather products (NWP). The authors detail the accuracy of each specific ML technique using all variables, expert model, and NWP forced regimes. They found the 'expert' models were highly effective and potentially more accessible for forecasting operations. Overall, the manuscript is generally well written and organized. The dataset appears suitable for this type of analysis, and the methods are appropriate and reasonable. However, there are several major topics the authors should consider. Here, I provide a few general comments and then specific comments below.*

- *1. The first two major issues pertain to data applicability and sample size reporting. The authors use data from ten winter seasons to train the data and then from five winters to test the data. Given non-stationarity of meteorological and climate variables as well as interannual variability, how well do the test data represent the training dataset and vice-versa. In other words, are you testing the model on winters similar enough to the winters on which they were trained? How might using time series data with the potential of non-stationarity influence your results? Can you provide some measure of variability and spread that shows these two winter datasets are reasonable for training and testing? Additionally, when assessing the 2019 season, was the season similar to the test data seasons? Given the decent scores, it appears so, However, when making recommendations for an operation, it is important to note that different variables or interactions of variables different than the training dataset could still result in avalanches resulting in poor model performance (Peitzsch et al., 2012).* We thank reviewer 2 for their helpful and constructive review. To provide a glimpse of the variability in our dataset, we added Figure 4-a, which shows the variability of event days and the total number of avalanches for each year, both for the *Train* and *Test* datasets. Figure 4-a also illustrates that maximum and minimum values are reached within the training dataset, and the *Test* dataset does not contain higher or lower values compared to the *Train* dataset. Additionally, the values for the winter of 2018-2019 are also within the average of both datasets.

- *2. You state the number of avalanches (n=861) over 153 event days from 1987 to 2020, but not for the period of record you studied. From the confusion matrix it looks like n=209 for the training dataset but n=489 for LR and NN, but n=479 for CT and RF for the test data (Table 6).* We provided a sample size for the *Train* and *Test* dataset in in the new Figure 4. As for the Table 6, a typo was removed so now n=479 for all ML methods.*1) Why are there over 2x as many event days in the test data compared to the training data when there are 2x as many years in the test data? This seems peculiar. If this is indeed the case, please provide background in the main text.* We clarified that the training dataset was balanced by randomly selecting the same number of avalanche event days. However, the *Test* dataset did not

use a balanced dataset approach; instead, it utilized a hindcast with all the days during the five winters, resulting in a larger sample size. 2) Why are there two different values (n=479 and n=489) for test data in Table 6? It would be helpful to state the sample size overall value of avalanches and event days for the study period, the sample size for $E_{AVA}$ during this period, and the sample size for the training and test datasets in the main text. We now state in Figure 4-a the number of event days and number of avalanches for the *Train* dataset and *Test* dataset.

*Additionally, the weighting for different number of interventions per day seems rather arbitrary.* In order to demonstrate our basis for the weight class of avalanche per day, we added Figure 4-b with the distribution of avalanche per days and the corresponding weighting bins. *Can you provide more context for using this metric as opposed to using perhaps a modification of the Avalanche Activity Index (Schweizer et al., 2003) that accounts for size. Accounting for size might also allow you to include avalanches impacting the road as well as those that don't. Forecasters are certainly considering avalanche activity that doesn't reach the object at risk and this might provide operations a continuum of results on which to base their decision.* Unfortunately, the avalanche size was not recorded in the dataset, so the avalanche activity index could not be used. We decided to use avalanches that reached the road as a significant event for our model. It was also the only consistent observations throughout a significant time in the dataset.

- *3. The comparison of ML output probability with Avalanche Québec's danger levels is a bit of a mismatch of scale. As you state, avalanche operations are incorporating more statistical and physically based snowpack models into their workflow. However, they are likely (hopefully) using scale appropriate simulations or datasets. In your study, the models are trained to predict avalanches that reach Road 132, a very specific local area.* We have now provided more details about the specific forecast of AvQc, which is very specific to avalanche occurrences on the 132 road.
*The danger ratings are a regional hazard forecast, not necessarily a true reflection of avalanche occurrence for that given day. It would be more useful to compare ML predictive capacity to avalanche occurrence (of similar size) throughout the region rather than danger ratings themselves, since the forecasted danger level also contains an unknown amount of uncertainty. Please discuss the rationale for choosing danger ratings and limitation therein.* We have now provided more details about the specific forecast of AvQc, which is particularly focused on avalanche occurrence on Road 132. With this specific forecast, we calculated the $F_1$ score for the *Test* dataset when AvQc's forecasted danger levels were *Considerable* and *High*. The ML predictive performance was compared primarily to avalanche occurrences in the study area using the $F_1$ and AUC scores. This comparison allows us to evaluate performance metrics ($F_1$ and AUC) between traditional forecasting and ML prediction. Figure 5 also shows avalanche predictions on the road compared to ML model predictions and avalanche forecasts. However, in Figure 5, we have visually minimized AvQc's forecast to put more emphasis on avalanche occurrences on the road.

- *4. There is currently very little to no mention of limitations and potential sources of uncertainty or error in the manuscript. Given your suggestion to use ML for operational forecasting workflows, it seems necessary to include a section that details the limitations in this study. It would also be useful to include potential considerations or statements of caution if avalanche forecasting operations are contemplating using ML model output for decision-making.* We added a new discussion section (5.3) about bias and uncertainty to address several points raised by all the reviewers, focusing mainly on our weather station's representativity and the potential overfitting of our duplication strategy.

- *5. As the authors point out, ML techniques have been used to study avalanche occurrence and improve avalanche forecasting. The use of these techniques is not novel. The addition of a comparison with a more parsimonious 'expert' model, the use of a NWP, and the integration of public avalanche forecasts for validation make this paper a worthy contribution to the existing body of literature. However, some of the Discussion reads like a consultant report for a specific client rather than a manuscript for the broader scientific community. For example, there are more suggestions for this specific forecasting operation than broad comparisons with other studies that might mike this manuscript more applicable to other operations. Please expand upon this in the Discussion. Additionally, there is a lot of content in the Discussion devoted to using the model in operational contexts. It would be helpful to mention how incorporating*

*these results/ML output differs from what the operation currently uses. In other words, how would things have improved (or not) if they used this model versus what they currently do? You allude to this in lines 373-375, but providing further evidence of how previous assessments would have been improved could be helpful here. It would also be useful to describe the days the models failed to predict events since the consequences would be more severe.* We thank the reviewer for this valuable comment. We have rewritten the entire section on operational implications (5.4) according to the recommendations above. The section now features less of a consulting report tone, enhanced comparisons with the forecasting performance of AvQc, more comparisons with existing studies, and additional recommendations for other operations.

**2  Specific and Technical Comments**

- *Abstract: please write/define all abbreviations (e.g. MTMQ, LR, and RF)* done

- *Line 24: spreading zone? Do you mean runout or deposition zone?* We changed it for "runout zone".

- *Line 29: 'windy' remove 'y'.,* We changed it for "wind-hardened snow".

- *Line 84/Figure 3: This is a bit confusing. The 'Data from NWP model' arrow makes it look like this was used to create the expert model variables. I thought the expert model variables were manually selected. Also, where is the expert model in this flow chart?* We changed the sentence for : "when using with numerical weather prediction (NWP) as input".

- *Line 85/Sec. 3.1: See comment above re: sample size values. It is necessary to know the sample size for training and test datasets in the text for proper evaluation.* The sample size is now shown in Figure 4, for both dataset.

- *Line 91: I don't understand how EAV A is a binary variable. It seems there are 4 possible values (1-4) used to weight if the binary value of avalanche hitting the road (1) or not (0) is 1. The variables simply is a weighting factor. Please clarify.* The sentence has been changed for : "A binary event day variable ($E_{AVA}$ = 'avalanche day' or 'no avalanche day') was created for days where an avalanche has reached the road, regardless of the size or path length because this information was not recorded (Table 1). If more than one intervention (avalanches) reached the road, duplicate event days were added to the dataset with the following weight of 1, 2, 3, or 4 duplicate days with respect to one intervention, two to five interventions, 6 to 9 interventions, and 10 or more interventions (Table 1)."

- *Line 109-110: The current wording is confusing because the probability of an avalanche reaching the road, as determined by the models, can also be less than 50%. Do you mean 'All probabilities above 50% were classified as event days.'? Also, change 'Considerate' to 'Considerable'.* The sentence was changed for : "With the forecast danger level of Avalanche Québec, we considered an event predicted with the *Considerable* level, where small avalanches are expected to reach the road . We also consider an event day where the probability of the ML mode is above 50%, enabling us to compare the ML algorithm probability to the Avalanche Quebec forecasted danger level (Considerable), as well as with the avalanche event dataset."

- *Line 140: BD should be DB, I assume, for database. Either define DB first or use full word.* BD was removed for dataset.

- *Line 171: Seems like you need a citation that demonstrates that model output can be daunting.* We changed the sentence for :" We used "expert model" to test the hypothesis that simpler non-processed meteorological variables could perform well in ML prediction for avalanche events in an operational avalanche management context."

- *Line 193: Change TC to CT.* done

- *Line 197: CA should be CT.* done

- *Line 212: 'data is processed' to 'data are processed'* done

- *Line 226: 'unbalanced datasets'. Since you balanced the training datasets (Sec. 3.1), I assume you mean 'unbalanced test datasets' as per the next sentence. Consider clarifying this here.* We changed for unbalanced test datasets.

- *Table 6,7,8: Please define acronyms for the reader in the caption. Also define Matrice (i.e. confusion matrix).* Done

- *Line 261: See general comment above re: representativeness of assessing model with just one season.* This section was modified based on the general comment above, it is now more clear that the models were evaluated with 5 years of hincast (Test dataset) and that the 2019 hindcast winter was a base for a comparison with the NWP prediction.

- *Line 314-319: It would be helpful to know the proportion of events classified as wet (wet loose, wet slab, or glide) in both the training and test datasets. You are simply including these variables because wet avalanches are likely to become more frequent, but the model is trained on historical data.* Unfortunately this avalanche type was not recorded by the observers.

- *Line 321: Use caution when suggesting practitioners can use specific threshold values for forecasting purposes. See general comment above regarding limitations. There is some uncertainty in the models and there should be some confidence intervals surrounding these values if recommending their use. Using specific model-derived values as a threshold for operational decision making can be problematic, particularly in the context of non-stationarity.* This line was removed and uncertainty is now addressed in a dedicated discussion section (5.3).

**References**

Peitzsch, E. H., Hendrikx, J., Fagre, D. B., and Reardon, B.: Examining spring wet slab and glide avalanche occurrence along the Going-to-the-Sun Road corridor, Glacier National Park, Montana, USA, Cold Regions Science and Technology, 78, 73–81, https://doi.org/10.1016/j.coldregions.2012.01.012, 2012.

Schweizer, J., Jamieson, J. B., and Schneebeli, M.: Snow avalanche formation, Reviews of Geophysics, 41, 3–5, https://doi.org/10.1029/2002RG000123, 2003.

---

## Author Comment (AC3)

**Response to reviewer's comments**

Manuscript EGUSPHERE-2025-1572

Assessing the predictive capability of several machine learning algorithms to forecast snow avalanches using numerical weather prediction model in eastern Canada

by Gauthier, Laliberté and Meloche

In the following, we provide (in blue) detailed point-by-point answers to the comments raised by the reviewers (in black, *italic*). In addition, modifications made to the manuscript are highlighted in a separate file with tracked-changes.

**Response to Referee #3– Dr. Cristina Pérez-Guillén**

*This paper presents several machine learning models for predicting avalanche events in eastern Canada. The study addresses an emergent topic in avalanche forecasting, with increasing demand for developing data-driven models that can be applied in practice. The authors used typical machine learning models, such as Logistic Regression (LC), Tree classification (TC), Random Forests (RF) model, and an Artificial Neural Network (NN), and compared the variation of the performance using variables extracted from meteorological data, local avalanche observations, and avalanche forecasts of Québec region. While the content, structure, and results of the paper are good and relevant to the topic, improvements are needed in the data used to develop the models, the definition of the target variable, and the detailed explanation of the methods and strategies employed. I recommend publishing this paper after the authors have addressed the following comments and suggestions. Please see my detailed feedback below.*

**1 General comments**

- *1. It would be very useful to add more detailed information on the types of avalanches typically released in the area, their locations (including typical paths in the map of Figure 1), and how these data and observations are collected. Are avalanches observed daily? Are the observations human-based? How accurately is the avalanche release time known? What happens during snowstorms? Is the road closed during periods of high avalanche danger? How far is the meteorological station from the avalanche release areas? Specifically, I recommend adding much more information about the type, size, and release time of the avalanche events in the dataset described in Section 3.1. Additionally, making this dataset open source would be beneficial for other studies.* The following lines were added to the section 3.1 to describe how the observations were collected :*The avalanche observations are collected by MTMQ patrollers to assess the road conditions for rock and ice falls, snow avalanches, as well as storm surge at all time. The observations are recorded at the hour of discovery of the debris, within a few hours of the actual avalanche. Even when the road is closed to the public, patrollers still moved along the road to assess road conditions and make observations. Then, the hourly events are sum to a daily dataset that contains the number of snow removal interventions on the road, the avalanche corridor, whether or not the road was reached, and the distance travelled (ditch, shoulder, 1 lane, 2 lanes). The type and avalanche size are not recorded by patrollers, and therefore, canoot be used in the analysis.'* The weather station is at the east of the avalanche release areas in Cap-Madeleine, indicated in Figure 1. We added these sentences : *"The weather station is located to the east of the study areas (Figure 1), at the sea level ( 0 m a.s.l). The altitude of the release areas in in average 50 m a.s.l, and goes up to 200 m in Mont-Saint-Pierre sector (Figure 1).* For the open-source comment, unfortunately the dataset belongs to the MTMQ who to not wan to make it open-source, but could be accessible upon request by researchers.

- *2. I suggest defining the target variable more clearly in Table 1 and the rest of the manuscript. For example, Eava = 'avalanche day' / 'no avalanche day'. This definition should be used consistently in all the confusion matrices of the tables. Also, to qualify as an avalanche day, what are the requirements? For example: at least one avalanche of size X (what is the minimum size or path length to reach the road?) Additionally, the selection of the weights in Table 1 is not justified or supported by previous studies.* The sentences in section 3.1 were restructured to describe more clearly the event variable :*A binary event day variable ($E_{AVA}$ = 'avalanche day' or 'no avalanche day') was created for days where an avalanche has*

*reached the road, regardless of the size or path length because this information was not recorded (Table 1). If more than one avalanche reached the road, duplicate event days were added to the dataset with the following weight of 1, 2, 3, or 4 duplicate days with respect to one intervention, two to five interventions, 6 to 9 interventions, and 10 or more interventions (Table 1).*" In addition, Figure 4b was added to justify the choice of the bin weight based on the frequency of avalanches per day.

- *3. Given that the meteorological drivers for dry and wet snow avalanches differ, are the models developed to predict dry avalanches, wet avalanches, or both? This is only mentioned in the Discussion section and could be earlier in the manuscript.* We added a sentence at the beginning of Data and Method section to state that the model will predict both dry and wet avalanches: " Our forecasting goal is to predict both dry and wet avalanches in a unique and simple daily model, which will predict the probability of an avalanche occurring". We also add a sentence in the method section stating that the type was not recorded so it couldn't be used in the analysis: "The type (dry or wet) and avalanche size are not recorded by the road patrol, and therefore, cannot be used in the analysis."

- *4. The explanation of how the data is merged and the target variables used to test the hindcast and forecast models can be explained in more detail. What is the time window of the avalanche forecasts from Avalanche Québec? How is this merged with the variables extracted from the time series of meteorological data and avalanche activity (point data)? The release time is very important when correlating with the meteorological variables used to develop the models. For instance, if an avalanche is released at 01:00, daily meteorological variables computed from 00:00 to 00:00 of the following day would not accurately represent that event.* The data is merged daily at midnight for each dataset and model. We added "(midnight)" after "daily" in the avalanche event section (3.1) and the meteorological data section (3.3). Unfortunately, this introduces some bias and uncertainty to our method, but it still shows efficient results with few misses, almost the same as traditional forecasting. We also believe that using weather variables derived from 48-hour and 72-hour periods helps reduce the error associated with using a midnight cutoff.

- *5. Could you please provide more information about the avalanche forecast issued in Québec? When is this forecast released, and what is the valid time window of the bulletin?* We added these sentences to describe the forecast: " The forecasts are issued everyday by Avalanche Québec's forecaster teams specifically for the avalanche paths along road 132. They based their prediction on a specific danger level scale relative to the occurrence of avalanches on these specific coastal paths along the road 132 (Figure 2)."

- *6. Why were these probability thresholds assigned to each danger level? It would be helpful to show the distribution of all the avalanche and non-avalanche events in relation to the danger level forecasts in Québec. This type of analysis could be used to justify or derive the chosen probability thresholds. Additionally, according to the European Avalanche Danger Scale, danger levels 1 and 2 do not imply the absence of avalanche activity and are dependent on avalanche size (1; 2). For instance, the time series results in Figure 4 show that 4 out of a total of 9 avalanches that reached the road occurred when the forecast danger level was 1 or 2.* We decided to remove entirely this section and only to present visually the model probability and the avalanche forecast. The danger scale is specific to the path along the coast and was decided by the forecaster and Quebec Ministry of Transportation.

- *7. Feature selection: It would be interesting to provide more information about the feature selection approach. For example, including a correlation matrix in the Appendix. Since Recursive Feature Elimination requires as input the number of input variables, it would be good to show how varying this number affects the performance of the LR and NN models. Does the final selection correspond to the number of features that yields the maximum F1-score?* We have now provided more details about our feature selection procedure for the LR and NN algorithms in subsection 3.4 'Learning Procedure'. Appendix A includes Table A1, which gives more details on the feature importance of each variable during the RFE-CV (feature selection procedure). We acknowledge that RFE is sensitive to the required number of input variables for the model, or how many features to keep. This is why we chose to perform RFE-CV, where the number of features to keep is defined by the maximum $F_1$ score. Then, we conduct a second RFE within each group to retain only one feature per group (best feature importance) to avoid collinearity in

the final variables. Thus, the final input variables will always be a maximum of 8 features, which corresponds to the number of correlated groups. Fewer than 8 will be the case if no feature within a correlated group was kept after the first RFE-CV.

- *8. Data splitting and balancing: You mention that splitting the data into training and test sets with a 70/30 ratio is optimal, and that the avalanche and no-avalanche events were balanced to train the models. It would be helpful to include, for example, a bar plot showing the total distribution of events for each class and their distribution in the training and test sets.* We added Figure 4-b to show the distribution of both datasets with each weight bins (number of avalanche per day). *The confusion matrices in Tables 6 and 7 do not reflect a balanced dataset. Please check my specific comments on these tables below.* The confusion matrices are balanced only for the *Train* dataset in Table 6. The *Test* dataset, as well as the hindcast for winter 2019 in Table 7, are not balanced to evaluate the predictive capability. Additionally, the confusion matrices have been slightly modified to correct typos in Tables 6 and 7.

- *9. I suggest moving the description of the machine learning models and the definition of the evaluation metrics to the Appendix. Instead, the main text could focus more on the final model selected, including details such as the architecture of the neural network and the hyperparameters used for the other models. Additionally, providing the code and data used to develop the models would improve the reproducibility of the study and be valuable for future research.* We added sentences to each machine learning description detailing the packages used, the hyperparameter values, and the specific architecture for each algorithm. However, we did not move this section to the appendix because we consider it an essential part of the methodology. The code will be available online, but the dataset belongs to Québec's Ministry of Transport, and we cannot publish it on their behalf as they are the data owners.

- *10. For evaluating the performance of the hindcast and forecast models (GEMLAM 24h and 48h) shown in Table 7, what "ground truth" is used to evaluate the performance? Are the danger level forecasts merged by combining levels 1 and 2 as "non-avalanche" events and levels 3 and 4 as "avalanche" events? This should be clearly stated in the caption of the tables and the main text. I am not convinced that merging danger levels 1 and 2 as non-avalanche and levels 3 and 4 as avalanche events is appropriate for defining the ground truth, especially since the actual ground truth in this case consists of observed avalanche events.* We are sorry that original manuscript was not enough clear on that matter. The ground truth in Table 7 is the avalanche event dataset, which was used for the computation of F1 and the AUC score. This information is now present in the caption. *Instead, presenting a correlation analysis between the model outputs and the avalanche forecasts may be more suitable.* The Avalanche forecast is now present as a visual comparison is presented in Figure 5, and the $F_1$ score of the AvQc forecast is computed with their *Considerable* or *High* danger level (occurence of avalanches on the road) with the avalanche event dataset.

**2   Specific comments**

- *Figure 1: It is difficult to see the location of the Cap-Madeleine weather station (the legend should also be translated to English). It would be very helpful to show the outlines or locations of the typical avalanche paths used in this study on the map.* The visibility of Cap-Madeline Weather station has been improved and the legend translated.

- *Lines 61–62: The references to Figure 1 are incorrect; they should refer to Figure 2.* Done

- *Tables 6 and 7:*

    - *The captions of these tables could be improved by including more detailed information.* The captions were improved and added more information for clarity.

    - *The differences between the results shown in the two tables are not clearly explained. Table 6 presents the performance of the models trained with a different set of variables after the feature selection process, right? Table 7 shows the performance of each model using only four variables. Why does the CT model use only three variables?* We improved the caption to state that Table 6 is

for the model selected by each ML models, and Table 7 for the expert models. We also add that, in Table 7, the CT algorithm did not selected the Rain_24h as significant when provided the 4 variables selected by the "expert".

– *The terms "train" and "train non double" are not defined in the main text or in the captions of the tables. Although it is mentioned that the models have been balanced (Line 158), the "train" dataset is not balanced, as shown in the confusion matrices of both Tables. Additionally, both "train" and "train non double" appear to contain significantly fewer samples than the test set. It is unclear whether the results for the test set were obtained using the "train" or "train double" datasets.* The train non double was removed completely for clarity. The Table now included only the train dataset (Balance) and the test (5 years of Hindcast).

– *The reason for showing the performance on these two training sets is also unclear, as their results are not discussed in the text. It is assumed that the results are based on 50 repetitions of the models (Line 160). If that is the case, are the reported scores the averages over those 50 repetitions? If so, a different confusion matrix should be obtained for each repetition.* The models were constructed 50 times per ML algorithm, thus creating 200 models for the train dataset. The best $F_1$ score is shown.

– *There is an inconsistency in the count of avalanche events reported in Table 6: the test set contains 42 events for the LR and NN models, while only 32 events for the other models. Also, the event counts in Table 7 for the NN model differ from the rest.* The typos in the count have been changed, now all the models count 32 events. LR and NN had 38 TP instead of 28 (Table 6). NN in Table 7 was also changed to get 32 events.

• *Table 8:*

– *The captions of these tables could be improved by including more detailed information.* The captions has been improved for clarity : "The performance metrics are computed with the avalanche event dataset where avalanches reached the road. An event day is considered when the danger level is Considerable or High (avalanche expected on the road), according to Table 2."

– *Which model is used here?* No model is used here but we compared the danger level (Considerable or High) of AvQc with avalanches on the road.

**References**